# Semantic-Aware Representation Learning for Lifelong Learning

**Fahad Sarfraz[1,2], Elahe Arani[1,3,*]& Bahram Zonooz[1,*]**

[1] Dep. of Mathematics and Computer Science, Eindhoven University of Technology (TU/e), Netherlands
[2] TomTom, Netherlands
[3] Wayve Technologies Ltd, London, United Kingdom
{f.sarfraz, e.arani, b.zonooz}@tue.nl

## Abstract

The human brain excels at lifelong learning by not only encoding information in sparse activation codes, but also leveraging rich semantic structures and relationships between newly encountered and previously learned objects. This ability to utilize semantic similarities is crucial for efficient learning and knowledge consolidation, yet it is often underutilized in current continual learning approaches. To bridge this gap, we propose *Semantic-Aware Representation Learning* (SARL), which employs sparse activations and a principled approach to evaluate similarities between objects encountered across different tasks and subsequently uses them to guide representation learning. Using these relationships, SARL enhances the reusability of features and reduces interference between tasks. This approach empowers the model to adapt to new information while maintaining stability, significantly improving performance in complex incremental learning scenarios. Our analysis demonstrates that SARL achieves a superior balance between plasticity and stability by harnessing the underlying semantic structure. [1]

## 1 Introduction

Deep neural networks (DNNs) have been designed for static in-distribution batch learning, whereby the model assumes that each batch of training data is representative of the underlying joint distribution. However, this assumption fails when processing sequential data in continual learning (CL) scenarios where information is made available incrementally over time, and hence the training batch at a given point only contains samples from the current task. The inadequacy of standard training to handle such dynamic data distributions often results in catastrophic forgetting (McCloskey & Cohen, 1989), erasing previously acquired knowledge as the model learns the new data. To address these challenges, CL methods aim to retain previous knowledge while learning new tasks, enabling models to accumulate knowledge across tasks (Parisi et al., 2019). The key challenge in CL is maintaining an optimal balance between stability and plasticity (Mermillod et al., 2013): the model needs enough plasticity to quickly adapt to new tasks, but also enough stability to preserve previously acquired knowledge. Existing approaches often trade off between high plasticity (quick adaptation) and high stability (knowledge retention), without effectively balancing the two. This trade-off highlights the difficulty in consolidating knowledge efficiently while learning sequentially.

Many CL approaches tend to emphasize either stability or plasticity, often overlooking the complex relationship between these two essential components. Stability-focused methods aim to preserve important weights to maintain previously acquired knowledge (Zenke et al., 2017; Li et al., 2023), while plasticity-oriented approaches allow significant representation changes to adapt flexibly to new tasks (Caccia et al., 2022; Liang & Li, 2023). This dichotomy often treats stability and plasticity as separate objectives rather than interconnected factors in knowledge consolidation. We hypothesize that leveraging semantic relationships between objects can bridge this gap by enhancing knowledge transfer and integrating new information with existing knowledge. By exploiting these semantic structures, a CL model can reuse and consolidate overlapping concepts, enabling it to learn new

---

[*]Equal advisory role.
[1]Code is available at https://github.com/NeurAI-Lab/SARL.git.

objects without severely disrupting previously learned representations. This approach mirrors the human cognitive ability to form associations between related concepts, facilitating retrieval, transfer, and consolidation of knowledge across tasks (Binder & Desai, 2011; Saxena et al., 2022).

The human brain exemplifies the power of semantic relationships in organizing and retaining knowledge. By forming associations between related concepts, the brain facilitates the retrieval, transfer, and consolidation of information. When encountering a new object similar to a previously learned one, the brain retrieves relevant information using semantic relationships (Saxena et al., 2022), aiding the learning of new objects and linking new information to existing knowledge. Furthermore, semantic relationships support knowledge consolidation and retention by reusing common concepts and adapting them in a cohesive manner. Moreover, semantic relationships play an important role in organizing memory (Artuso et al., 2022), necessitating that information be encoded in a way that captures these relationships. The brain employs sparse coding (Foldiak, 2003), where information is represented by the strong activation of a small subset of neurons. Notably, objects sharing concepts likely activate overlapping sets of neurons, efficiently encoding semantic relationships. We hypothesize that emulating these mechanisms can enable DNNs to better balance stability and plasticity by leveraging the rich semantic structure between objects.

To this end, Semantic-Aware Representation Learning (SARL) emulates sparse coding in the brain and employs a principled approach to capture the semantic relationships between objects and to use them to guide representation learning. SARL leverages activation sparsity to emulate sparse coding and represents each object through semantically rich object prototypes (Snell et al., 2017) derived from the mean sparse activations of the object samples. These prototypes effectively capture the semantic structure of object relationships, allowing the model to reuse and consolidate knowledge efficiently. By capturing the inherent similarities between objects, SARL leverages semantic relationships to guide the alignment of new object prototypes with those of similar classes from previously learned tasks, while promoting separation from dissimilar ones. This alignment encourages a structured representation of knowledge, facilitating knowledge transfer across tasks and reducing interference. Furthermore, SARL incorporates prototype regularization to ensure model stability, mitigating the risk of forgetting previously learned information, and fostering effective consolidation of knowledge in a cohesive manner. This mechanism enables the preservation of valuable representations while adapting to new tasks, ultimately enhancing the model's overall learning capabilities.

Our empirical analysis on challenging class-incremental learning scenarios across various datasets demonstrates that SARL significantly enhances lifelong learning performance by leveraging semantic structure for representation learning. By effectively aligning representations based on semantic relationships, SARL facilitates efficient knowledge transfer and consolidation, enabling the model to adapt seamlessly to new tasks while retaining previously acquired knowledge. These capabilities are further supported by our analysis, which reveals that SARL achieves a better balance between model stability and plasticity, mitigates forgetting, and reduces task recency bias.

## 2 RELATED WORK

Approaches to addressing catastrophic forgetting in CL can be broadly categorized into three groups. *Regularization-based* methods penalize changes in the model either in parameter space (Farajtabar et al., 2020; Ritter et al., 2018) or functional space (Benjamin et al., 2019; Li & Hoiem, 2017). These methods primarily focus on stability and generally fail in class-incremental learning settings where task information is unavailable. *Dynamic architecture* approaches (Rusu et al., 2016), expand the network to allocate distinct parameters for each task. While reducing forgetting, these methods lead to model size scaling linearly with the number of tasks and often require task identity at test time, limiting their applicability. *Rehearsal-based* approaches draw inspiration from the brain's experience replay for knowledge consolidation (Ólafsdóttir et al., 2018). They store a subset of data samples from previous tasks in a memory buffer, which are interleaved with new tasks samples to approximate the joint distribution. Rehearsal-based approaches have proven to be more general and effective for various continual learning scenarios (Farquhar & Gal, 2018).

The baseline Experience Replay (ER) (Riemer et al., 2018) interleaves the training of new task samples with previous task samples in memory. Several approaches have since been proposed to provide additional learning cues to the model from its previous state. Dark Experience Replay (DER++)

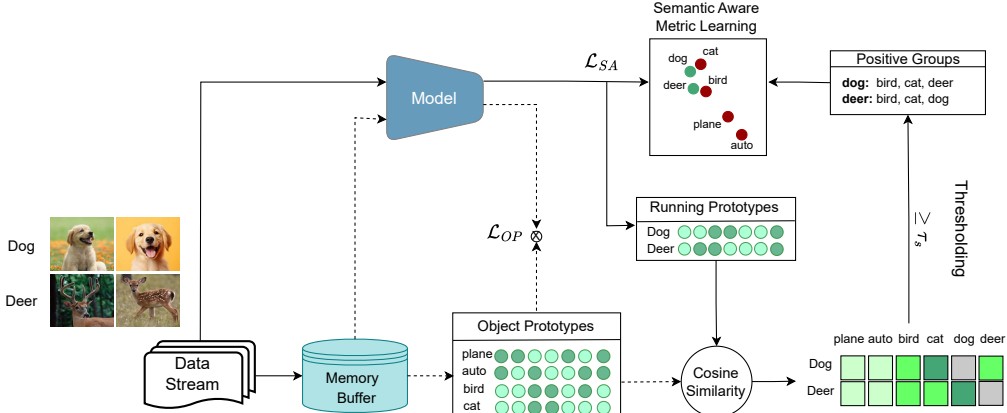

Figure 1: SARL employs activation sparsity to emulate brain-like sparse coding, representing each object with a class prototype derived from the mean representations of object samples. The semantic relationships are utilized to encourage new object prototypes to align with the class prototypes of similar objects and diverge from dissimilar ones. Additionally, SARL ensures model stability through prototype regularization, mitigating forgetting and enabling effective consolidation of information in lifelong learning. Darker shades represent higher values.

(Buzzega et al., 2020) additionally stores the output logits and applies a consistency loss to the memory samples. ER-ACE (Caccia et al., 2022) uses an asymmetric cross-entropy loss, considering only new task logits, on the incoming samples to reduce representation drift. Gradient Coreset Replay (GCR)(Tiwari et al., 2022) selects and maintains a core-set based on gradient approximation.

Recently, there has been a shift towards multi-memory-based CL approaches, inspired by the complementary learning systems theory in the brain (McClelland et al., 1995). CLS-ER (Arani et al., 2022) simulates the interaction between fast and slow learning by using two semantic memories that aggregate model weights at different rates with an exponential moving average. $CO^2L$ (Cha et al., 2021) initially learns representations with a modified SupCon loss and subsequently trains a classifier using samples from the last task and buffer data. Deep Retrieval and Imagination (DRI)(Wang et al., 2022a) leverages an embedding network and generative model to retrieve and generate imaginary data. SCoMMER (Sarfraz et al., 2023) enforces sparse coding for efficient representation learning and uses multiple memories, using the previous state of the model to reduce similarity drift. While these multi-memory approaches effectively reduce forgetting, they incur additional computational and memory costs, which can be prohibitive in some real-world settings. Therefore, we believe that it is crucial to improve CL performance in a single-model setting, which can later be extended to a dual-memory framework. Please see extended related work in Appendix G.

## 3 METHODOLOGY

We begin with an overview of the biological mechanisms that inspire our approach, followed by details on how we mimic these mechanisms in DNNs and an outline of our overall formulation.

### 3.1 BIOLOGICAL MOTIVATION

The human brain excels at adapting to dynamic environments by transferring knowledge from previously learned concepts to new ones, mitigating interference, and efficiently consolidating information. A critical component of this lifelong learning ability is the brain's capacity to exploit semantic relationships between objects, which guides both information encoding and memory formation (Binder & Desai, 2011). By establishing associations between related concepts, the brain organizes information into semantic networks in which interconnected concepts share common features, enabling efficient retrieval, transfer, and consolidation of knowledge (Saxena et al., 2022). When encountering a new objects, the brain rapidly integrates them into these networks, leveraging previ-

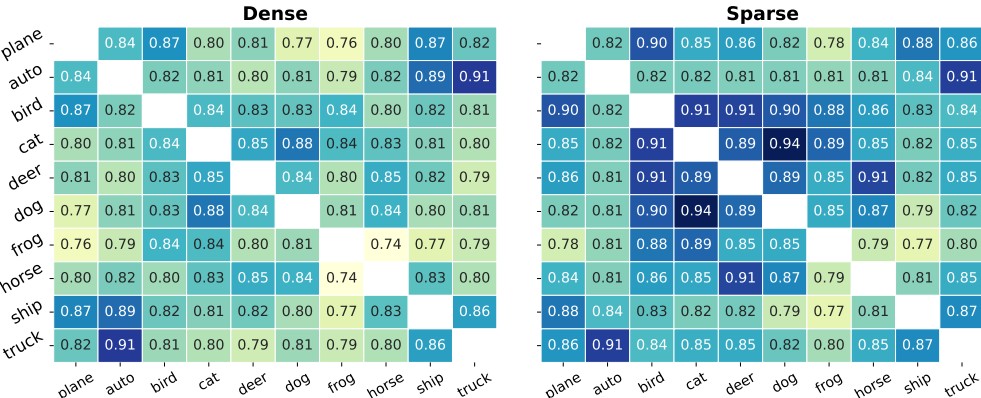

Figure 2: Similarity matrices were computed using object representations from the model trained on the joint CIFAR-10 dataset, with dense and sparse activations, respectively.

ously acquired knowledge to facilitate learning and reduce relearning efforts. This process is further supported by the brain's use of *sparse coding* (Foldiak, 2003), where only a small subset of neurons is activated. Objects with conceptual similarities activate overlapping neuron sets, hence efficiently encoding semantic relationships which can be leveraged for retrieval and learning.

Inspired by these cognitive mechanisms, we propose that incorporating sparsely activated neurons, which allow for the extraction of semantic structures, and leveraging these relationships for learning in DNNs, can significantly enhance their lifelong learning capabilities.

## 3.2 SEMANTIC-AWARE REPRESENTATION LEARNING

Building on these cognitive insights, we introduce Semantic-Aware Representation Learning (SARL), which adopts a principled approach to extract the semantic relationships between objects encountered across sequential tasks and subsequently utilizes them to guide representation learning in a cohesive manner. SARL draws inspiration from the brain's sparse coding approach and employs activation sparsity to facilitate semantic information encoding in representations. Each object is represented by an object prototype, capturing the average features of its instances, which are saved in memory at the end of each task. By evaluating the similarity between these stored prototypes and the average features of new objects, SARL identifies sets of positive (similar) and negative (dissimilar) objects from both previous and current tasks. Leveraging these relationships, SARL applies a novel semantic-aware metric learning approach, guiding the model to bring representations of new objects closer to similar prototypes while distancing them from dissimilar ones. This mechanism enables the model to reuse knowledge from similar objects, enhancing learning efficiency and reducing interference. To further promote stability, SARL implements prototype-based regularization, encouraging the activations of previously learned objects to remain consistent with their established prototypes and thus preventing drift over time. By integrating new information in a structured manner, SARL fosters effective knowledge consolidation and maintains a balance between plasticity and stability in lifelong learning. The following sections introduce the different components of SARL.

## 3.3 SPARSE CODING

To mimic sparse coding in the brain, we employ activation sparsity using k-winners-take-all (k-WTA) (Maass, 2000), where only the K most active neurons in each layer are allowed to propagate to the next layer. For convolutional layers, we apply k-WTA locally across the channels at each spatial location rather than globally across the entire input and all channels. This localized approach ensures that the most relevant features are selected at each position, promoting specialization of filters. As a result, the network can focus on distinct patterns at different spatial locations, leading to more efficient feature extraction, better semantic encoding, and reduced interference in lifelong learning scenarios.

### 3.3.1 OBJECT PROTOTYPES

Building on the sparse coding mechanism through k-WTA, we extend this principle to our formulation of object prototypes. These prototypes are central to capturing and incorporating the intrinsic similarities between objects to guide representation learning. By capturing shared concepts through overlapping sparse activations, k-WTA offers an efficient and intuitive method to encode such relationships. Each object is represented by its object prototype, defined as the mean of all instance representations, a strategy commonly employed in metric learning (Kaya & Bilge, 2019). Specifically, we utilize the $\ell_2$-normalized activations of the penultimate layer as object representations, with the prototype for each class $c$ computed as:

$$\mathcal{O}_c = \frac{1}{N_c} \sum_i \mathbb{I}(y_i = c) a'_i \tag{1}$$

where $a'_i$ represents the normalized activation of sample $x_i$ with label $y_i$, and $N_c$ is the number of instances in class $c$. This provides an efficient means of forming object prototypes that guide subsequent learning. At the end of each task, we calculate and store the object prototypes of all the new object categories. Hence, while learning task $t$, we have access to the object prototypes for all the object categories learned until task $t-1$.

### 3.3.2 SEMANTIC-AWARE METRIC LEARNING

Having developed an efficient method for representing objects using prototypes that capture the essence of each category, we can now evaluate the similarity between objects across tasks. We use cosine similarity, which measures the cosine of the angle between two class prototypes, providing a scale-invariant measure independent of magnitude. This approach is particularly advantageous when comparing object prototypes, as it focuses on directional alignment between vectors rather than their magnitudes, effectively capturing semantic similarity in high-dimensional spaces. The similarity between the object categories $i$ and $j$, and the corresponding object prototypes $\mathcal{O}_i$ and $\mathcal{O}_j$ is given by:

$$sim(\mathcal{O}_i, \mathcal{O}_j) = \frac{\mathcal{O}_i \cdot \mathcal{O}_j}{\|\mathcal{O}_i\| \cdot \|\mathcal{O}_j\|} \tag{2}$$

This provides us with an efficient approach to evaluating the similarity between object categories encountered across different tasks, which is subsequently used to inform learning. Figure 2 shows that the proposed approach can capture semantic similarities, and sparse activations further enhance the model's ability to distinguish between similar and dissimilar objects. Section E in Appendix shows that sparse activations enhances the capacity of the model to capture semantic similarities between objects within animal and vehicle clusters as well as between the two clusters.

Specifically, for each new object category introduced in task $t$, we calculate intermediate object prototypes $\overline{\mathcal{O}}$ using Equation 1, and evaluate the similarity between each new object and other objects in the combined set of both newly introduced and previously learned object categories. We employ a warm-up stage for each new task to allow the model to learn meaningful representations before computing the intermediate object prototypes. For each new object category $c$, we use the degree of similarity between object prototypes to identify a set of similar object categories $\mathbb{S}_c$:

$$\mathbb{S}_c = \{i \mid \mathcal{O}_i \geq \tau_s \wedge i \neq c \quad \forall i \in \mathbb{C}^{0:t}\} \tag{3}$$

where $\tau_s$ is the similarity threshold ranging from 0 to 1, and $\mathbb{C}^{0:t}$ represents the set of all object categories observed up to task $t$.

We employ our semantic-aware metric learning loss in subsequent epochs following the warm-up stage. For each new object category $c$, we encourage the mean $\ell_2$-normalized representations of its object instances in the batch, denoted as $\overline{o_c}$, to be closer to the object prototypes of objects in the set $\mathbb{S}_c$—indicating similarity. Simultaneously, they are pushed to be distant from dissimilar objects. Formally, the loss is given by:

$$\mathcal{L}_{SM} = \sum_{c \in \mathbb{C}^t} \frac{\sum_{i \in \mathbb{S}_c} \|\mathcal{O}_i - \overline{o}_c\|^2}{\sum_{i \notin \mathbb{S}_c} \|\mathcal{O}_i - \overline{o}_c\|^2} \tag{4}$$

where $\mathbb{C}^t$ is the set of new object categories at task $t$. The semantic-aware metric learning loss enables the model to leverage similarities between objects and acquire semantic-aware representations. This, in turn, facilitates effective knowledge sharing and reusing while mitigating interference. Such enhancements augment the model's capacity to consolidate knowledge more cohesively.

### 3.3.3 REGULARIZATION ON OBJECT PROTOTYPES

To enhance stability and preserve the semantic structure within the acquired representations, we introduce a regularization loss on the object prototypes. This regularization is applied to buffer samples, penalizing the divergence of mean representations of buffered samples in the batch from their originally learned object prototypes. Formally, in each training batch, we sample from the memory buffer, $M_b$, calculate the $\ell_2$-normalized mean representations for each object category, $o_c^b$, and minimize the mean squared error between $o_c^b$ and their corresponding object prototypes, $\mathcal{O}_c$, stored in the prototype memory, $M_o$:

$$\mathcal{L}_{OP} = \sum_{c \in \mathbb{C}^{0:t-1}} \|o_c^b - \mathcal{O}_c\|^2 \tag{5}$$

where $\mathbb{C}^{0:t-1}$ is the set of previously learned object categories. By imposing this regularization, we encourage the model to maintain the semantic integrity of learned object categories across tasks, minimizing deviations in the representations of buffered samples from their learned prototypes. This not only fosters stability in the model's knowledge but also mitigates the risk of forgetting important semantic relationships established in earlier learning phases.

### 3.3.4 FUNCTIONAL REGULARIZATION

Functional regularization plays a crucial role in helping models retain previous knowledge and effectively consolidate new information by directly regulating changes in the model's input-output behavior rather than just its parameters. When applied to new data, it encourages the model to adapt while remaining within the functional vicinity of its previous state, thereby facilitating the smooth integration of new information. On old data, functional regularization helps maintain consistency with prior learned functions, minimizing forgetting by directly aligning the current model's outputs with those from earlier tasks. This approach stabilizes the learning process and allows the model to evolve while preserving essential knowledge.

Specifically, before training on a new task, we pass its samples through the model and save the output logits, denoted as $z^t$, in memory. This allows us to enforce functional consistency during training on the new task without the need to maintain the previous model state in memory. To further promote consistency in model behavior and maintain semantic relationships, we also save the output logits along with the data samples in a memory buffer. During the replay of these samples, similar to Buzzega et al. (2020), we apply a consistency regularization loss on the model's output. At each training step with new task samples $x^t$ and $y^t$, along with their precomputed output logits $z^t$ from previous model state, we sample $x^b, y^b$, and $z^b$ from the memory buffer and apply the following functional consistency loss:

$$\mathcal{L}_{FR} = \alpha \|F(x^b : \theta) - z^b\|^2 + \beta |F(x^t : \theta) - z^t\|^2 \tag{6}$$

where $F(.)$ is the model's output parameterized by weights $\theta$. $\alpha$ and $\beta$ control the strength of functional consistency on buffer samples and new task samples, respectively. This combination of regularization on object prototypes and regularization of consistency improves the stability of the model and preserves the semantic structure learned in previous tasks.

### 3.4 OVERALL FORMULATION

SARL involves training a model $f(.; \theta)$ on a non static data stream, $D$ containing a sequence of T i.i.d tasks $(D^1, D^2, .., D^T)$. To employ rehearsal, it maintains a small episodic memory, $M_b$ using reservoir sampling (Vitter, 1985). Additionally, it maintains a memory of object prototypes, $M_o$. During each training iteration of task t, the model is fed with the training batch $(x^t, y^t) \sim D^t$ and a random batch from episodic memory $(x^b, y^b, z^b) \sim M_b$. Task 1 is trained using standard cross-entropy loss. For each subsequent task, we employ a warm-up stage where we train the model

Table 1: Comparison analysis of single- and dual-model CL methods across various CL settings. The baseline taken from the repective methods. We ran SCoMMER for Seq-TinyImageNet using the hyperparameter setting approach provided by the authors. We report the average accuracy and 1 std of 3 different seeds.

| Buffer | Method | Seq-CIFAR10 | | Seq-CIFAR100 | | Seq-TinyImg | |
|---|---|---|---|---|---|---|---|
| | | Class-IL | Task-IL | Class-IL | Task-IL | Class-IL | Task-IL |
| – | JOINT | $92.20_{\pm0.15}$ | $98.31_{\pm0.12}$ | $70.62_{\pm0.64}$ | $86.19_{\pm0.43}$ | $59.99_{\pm0.19}$ | $82.04_{\pm0.10}$ |
| | SGD | $19.62_{\pm0.05}$ | $61.02_{\pm3.33}$ | $17.58_{\pm0.04}$ | $40.46_{\pm0.99}$ | $7.92_{\pm0.26}$ | $18.31_{\pm0.68}$ |
| 200 | ER | $44.79_{\pm1.86}$ | $91.19_{\pm0.94}$ | $21.40_{\pm0.22}$ | $61.36_{\pm0.39}$ | $8.49_{\pm0.16}$ | $38.17_{\pm2.00}$ |
| | FDR | $30.91_{\pm2.74}$ | $91.01_{\pm2.74}$ | $22.02_{\pm0.008}$ | $61.72_{\pm1.02}$ | $8.70_{\pm0.19}$ | $40.36_{\pm0.68}$ |
| | DER++ | $64.88_{\pm1.17}$ | $91.92_{\pm0.60}$ | $29.60_{\pm1.14}$ | $62.49_{\pm0.78}$ | $10.96_{\pm1.17}$ | $40.87_{\pm1.16}$ |
| | ER-ACE | $62.08_{\pm1.44}$ | $35.17_{\pm1.17}$ | $27.44_{\pm0.64}$ | $25.29_{\pm1.89}$ | $11.25_{\pm0.54}$ | $44.17_{\pm1.02}$ |
| | GCR | $64.84_{\pm1.63}$ | $90.8_{\pm1.05}$ | $33.69_{\pm1.40}$ | $64.24_{\pm0.83}$ | $13.05_{\pm0.91}$ | $42.11_{\pm1.01}$ |
| | CO$^2$L | $65.57_{\pm1.37}$ | $93.43_{\pm0.78}$ | $31.90_{\pm0.38}$ | $55.02_{\pm0.3}$ | $13.88_{\pm0.40}$ | $42.37_{\pm0.74}$ |
| | CLS-ER | $66.19_{\pm0.75}$ | $93.90_{\pm0.60}$ | $43.80_{\pm1.89}$ | $73.49_{\pm1.04}$ | $23.47_{\pm0.80}$ | $49.60_{\pm0.72}$ |
| | SCoMMER | $69.19_{\pm0.61}$ | $93.20_{\pm0.10}$ | $40.25_{\pm0.05}$ | $69.39_{\pm0.43}$ | $16.30_{\pm1.54}$ | $49.37_{\pm0.81}$ |
| | **SARL** | $\mathbf{70.97}_{\pm0.47}$ | $\mathbf{95.72}_{\pm0.36}$ | $\mathbf{48.96}_{\pm0.53}$ | $\mathbf{78.91}_{\pm0.15}$ | $\mathbf{28.95}_{\pm1.13}$ | $\mathbf{71.56}_{\pm0.18}$ |
| 500 | ER | $57.74_{\pm0.27}$ | $93.61_{\pm0.27}$ | $28.02_{\pm0.31}$ | $68.23_{\pm0.16}$ | $9.99_{\pm0.29}$ | $48.64_{\pm0.46}$ |
| | FDR | $28.71_{\pm3.23}$ | $93.29_{\pm0.59}$ | $29.19_{\pm0.33}$ | $69.76_{\pm0.51}$ | $10.54_{\pm0.21}$ | $49.88_{\pm0.71}$ |
| | DER++ | $72.70_{\pm1.36}$ | $93.88_{\pm0.50}$ | $41.40_{\pm0.96}$ | $70.61_{\pm0.11}$ | $19.38_{\pm1.41}$ | $51.91_{\pm0.68}$ |
| | ER-ACE | $68.45_{\pm1.78}$ | $40.67_{\pm0.06}$ | $30.14_{\pm1.11}$ | $24.81_{\pm0.63}$ | $17.73_{\pm0.56}$ | $49.99_{\pm1.51}$ |
| | GCR | $74.69_{\pm0.85}$ | $94.44_{\pm0.32}$ | $45.91_{\pm1.30}$ | $71.64_{\pm2.10}$ | $19.66_{\pm0.68}$ | $52.99_{\pm0.89}$ |
| | CO$^2$L | $74.26_{\pm0.77}$ | $39.21_{\pm0.39}$ | $39.21_{\pm0.39}$ | $62.98_{\pm0.58}$ | $20.12_{\pm0.42}$ | $53.04_{\pm0.69}$ |
| | CLS-ER | $75.22_{\pm0.71}$ | $94.94_{\pm0.53}$ | $51.40_{\pm1.00}$ | $78.12_{\pm0.24}$ | $31.03_{\pm0.56}$ | $60.41_{\pm0.50}$ |
| | SCoMMER | $74.97_{\pm1.05}$ | $94.36_{\pm0.06}$ | $49.63_{\pm1.43}$ | $75.49_{\pm0.43}$ | $22.60_{\pm0.85}$ | $58.46_{\pm1.31}$ |
| | **SARL** | $\mathbf{75.64}_{\pm0.36}$ | $\mathbf{95.95}_{\pm0.24}$ | $\mathbf{55.30}_{\pm0.61}$ | $\mathbf{81.23}_{\pm0.24}$ | $\mathbf{32.56}_{\pm1.23}$ | $\mathbf{70.94}_{\pm0.43}$ |

with the combination of cross-entropy loss, $\mathcal{L}_{CE}$ on the training batch and buffer samples and the regularization losses. At the end of the warm-up stage when the model has learned meaningful representations, we evaluate the intermediate object references for each object category, c, in the current task using Equation 1 and identify a set of corresponding similar object categories, $\mathbb{S}_c$ using Equation 3 to employ the semantic-aware metric learning loss in the subsequent training on the task. The overall loss for training the model is given by:

$$\mathcal{L} = \mathcal{L}_{CE}(F(x^t : \theta), y^t) + \mathcal{L}_{CE}(F(x^b : \theta), y^b) + \lambda_{SM}\mathcal{L}_{SM} + \lambda_{OP}\mathcal{L}_{OP} + \mathcal{L}_{FR} \qquad (7)$$

where $\lambda_{SM}$ and $\lambda_{OP}$ are the hyperparameters that control the weight of each loss.

At the end of each task, we evaluate the object prototypes of the object categories introduced in the task using Eqn 1. Additionally, to mitigate the biases of the batch norm statistics towards the current task (Pham et al., 2022), we employ a simple yet effective technique of passing the model through all buffer samples to shift the batch norm statistics towards the approximate joint distribution.

## 4 EMPIRICAL EVALUATION

**Settings.** To evaluate the effectiveness of SARL, we consider various CL scenarios that each present unique challenges for lifelong learning models. In the *class-incremental learning (Class-IL)* scenario, with each new task, the model is introduced to a new set of classes and must learn to distinguish not only among the current task's classes but also across all previously encountered classes. This setting tests the model's ability to build generalizable representations, consolidate knowledge, and transfer that knowledge to efficiently distinguish all classes seen so far. We only train our models for the CLass-IL setting; for completion, we also provide *task-incremental learning (Task-IL)* setting. In this context, the model has access to task labels during inference, enabling the utilization of task labels to selectively choose the subset of output logits.

While Class-IL focuses on knowledge accumulation across distinct tasks, it does not fully capture the complexity of real-world scenarios, where tasks often lack clear boundaries, and classes can

Table 2: Comparison analysis of single- and dual-model CL methods on GCIL-CIFAR-100 dataset.

| Dist. | Buffer | JOINT | SGD | ER | DER++ | CLS-ER | SCoMMER | SARL |
|---|---|---|---|---|---|---|---|---|
| Uniform | 200 | $58.36_{\pm1.02}$ | $12.67_{\pm0.24}$ | $16.40_{\pm0.37}$ | $18.84_{\pm0.60}$ | $25.06_{\pm0.81}$ | $30.84_{\pm0.80}$ | $\mathbf{36.97}_{\pm0.54}$ |
| | 500 | | | $28.21_{\pm0.69}$ | $32.92_{\pm0.74}$ | $36.34_{\pm0.59}$ | $36.87_{\pm0.36}$ | $\mathbf{39.03}_{\pm0.45}$ |
| Longtail | 200 | $56.94_{\pm1.56}$ | $22.88_{\pm0.53}$ | $19.27_{\pm0.77}$ | $26.94_{\pm1.27}$ | $28.54_{\pm0.87}$ | $29.08_{\pm0.31}$ | $\mathbf{35.26}_{\pm0.75}$ |
| | 500 | | | $20.30_{\pm0.63}$ | $25.82_{\pm0.83}$ | $28.63_{\pm0.68}$ | $35.20_{\pm0.21}$ | $\mathbf{35.47}_{\pm1.20}$ |

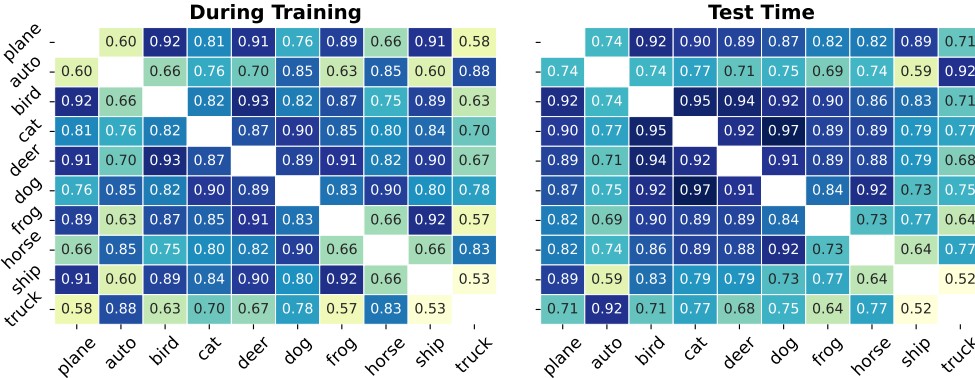

Figure 3: Cosine similarities between the object prototypes computed at the end of each training task on Seq-CIFAR10 with 200 buffer size and the similarity structure of the final model on test set.

reappear with varying distributions. To address this, we also evaluate SARL in the *generalized class-incremental learning (GCIL)* setting. GCIL introduces a more dynamic and realistic environment where classes reoccur across tasks, and the number of classes and their sample sizes can vary. This scenario tests the model's ability to deal with class imbalance, sample efficiency, and the continual integration of knowledge from overlapping or reappearing classes.

Additionally, we consider online continual learning (Online CL), a highly challenging setting where the model is exposed to each data instance only once, requiring it to learn and adapt dynamically without revisiting previous data. Details of the experimental setup and datasets are provided in Appendix A.

**Results.** We benchmarked SARL's performance against established rehearsal-based approaches across varying CL settings and dataset complexities. Table 1 demonstrates consistent performance improvements across all settings. Notably, under the low buffer regime, where the model has to retain knowledge while having access to very limited samples from previous tasks (approximately two samples per class for Seq-CIFAR-100 and one sample per class for Seq-Tiny-ImageNet), SARL effectively retains previously acquired knowledge, demonstrating its ability to learn new tasks while remaining in the functional vicinity of earlier tasks despite the limited buffer size and consolidating knowledge as it learns new tasks. Interestingly, as the complexity of the dataset increases with the number of classes with high semantic similarity, the performance gains are more pronounced, suggesting that SARL is able to exploit the semantic structure and utilize it to consolidate knowledge.

Table 2 highlights the advantage of SARL in the challenging GCIL setting, where the model must update its knowledge across multiple occurrences of the same object with varying sample sizes, while addressing class imbalance and task complexity. SARL's ability to update object prototypes when re-encountering a class ensures efficient use of new samples without penalizing for deviations from earlier suboptimal representations. SARL shows remarkable resilience to these challenges and provides additional credence to the benefits of semantic-aware representation learning.

These results are particularly impressive considering that SARL uses only a single model, while many state-of-the-art methods rely on multiple models. For instance, CLS-ER uses two additional models, and SCoMMER incorporates one. Although these methods claim to use a single model for inference, the distinction between training and inference in lifelong learning is blurred, as the model

Table 3: Performance comparison across Seq-CIFAR10 and Seq-CIFAR100 in the online CL setting. Baseline results are taken from Wu et al. (2024). We report the mean and std of three runs.

| Method | Seq-CIFAR10 | | Seq-CIFAR100 | |
|---|---|---|---|---|
| | AAA | **Acc** | AAA | Acc |
| SGD | $34.85_{\pm1.71}$ | $16.96_{\pm0.60}$ | $11.63_{\pm0.38}$ | $5.27_{\pm0.28}$ |
| ER | $55.53_{\pm2.58}$ | $43.83_{\pm4.84}$ | $23.19_{\pm0.38}$ | $16.07_{\pm0.88}$ |
| DER | $45.85_{\pm1.62}$ | $29.87_{\pm2.95}$ | $13.35_{\pm0.36}$ | $6.12_{\pm0.18}$ |
| DER++ | $64.22_{\pm0.70}$ | $52.29_{\pm1.86}$ | $19.88_{\pm0.43}$ | $11.79_{\pm0.65}$ |
| CLSER | $63.02_{\pm1.54}$ | $52.80_{\pm1.66}$ | $25.46_{\pm0.57}$ | $17.88_{\pm0.69}$ |
| OCM | $66.14_{\pm0.95}$ | $53.39_{\pm1.00}$ | $22.54_{\pm0.79}$ | $14.40_{\pm0.82}$ |
| ER-OBC | $65.82_{\pm0.91}$ | $54.85_{\pm2.16}$ | $25.54_{\pm0.25}$ | $17.21_{\pm0.92}$ |
| On-EWC | $38.44_{\pm0.50}$ | $17.12_{\pm0.51}$ | $11.81_{\pm0.42}$ | $5.88_{\pm0.31}$ |
| IS | $37.33_{\pm0.23}$ | $17.39_{\pm0.19}$ | $12.32_{\pm0.22}$ | $5.20_{\pm0.18}$ |
| La-MAML | $42.98_{\pm1.60}$ | $33.43_{\pm1.21}$ | $12.55_{\pm0.39}$ | $11.78_{\pm0.65}$ |
| VR-MCL | $\mathbf{66.97}_{\pm1.58}$ | $56.48_{\pm1.79}$ | $27.01_{\pm0.48}$ | $19.49_{\pm0.69}$ |
| SARL | $66.85_{\pm1.15}$ | $\mathbf{57.21}_{\pm0.27}$ | $\mathbf{31.66}_{\pm1.49}$ | $\mathbf{24.39}_{\pm1.44}$ |

must continuously learn from the environment. This can make the overhead of maintaining multiple memories prohibitive in real-world applications, especially for resource-constrained systems like embedded devices. The ability of SARL to achieve superior results with a single model further underscores its practicality and efficiency for lifelong learning.

Additionally, while SARL is not explicitly designed for Online CL, we adapted it to this setting to assess its versatility and broader applicability. By maintaining running feature sums and sample counts to dynamically update object prototypes, SARL effectively learns in an online manner. We introduce a brief warm-up phase during the first 10 iterations of each task to stabilize feature accumulation before evaluating prototypes. Despite this adaptation being secondary to our primary focus, SARL-Online demonstrates strong performance. As shown in Table 3, it achieves notable improvements across all datasets, particularly on Seq-CIFAR100, where it significantly boosts both **AAA** and **Acc**. These results further reinforce the effectiveness of semantic-aware representation learning, highlighting SARL's adaptability to different continual learning paradigms without requiring extensive modifications. Our ablation study in Appendix B.1 shows that each component of SARL contributes to the performance gains.

## 5 ANALYSIS

Here, we delve deeper into what enables performance improvements in SARL. We compare with the baseline ER and the state-of-the-art in single-model (DER++) and multiple-memory (SCoMMER).

### 5.1 SEMANTIC STRUCTURE

At the core of SARL is semantic-aware representation learning. We investigate whether SARL can learn and retain a meaningful semantic structure during sequential task training. We evaluate the similarities between object prototypes calculated stored in memory during training that are evaluated at the end of each task for new object categories and compared them with the similarity structure of the trained model on test data. Figure 3 shows that SARL not only retains the similarity structure but also updates it by consolidating new knowledge. Notably, the class prototypes from early tasks, where the model was trained to distinguish only between two classes, show strong similarities even with other classes, indicating that the initial representations were not yet fully optimized for all classes. However, as the model encounters more tasks, the semantic relationships across classes become more refined and meaningful. For instance, classes such as 'cat', 'dog', and 'horse' exhibit higher similarity values post-training, reflecting their semantic coherence, while unrelated classes such as 'plane' and 'truck' maintain lower similarities. The semantic-aware metric learning loss, combined with object prototype regularization, guides training to enforce and maintain this semantic

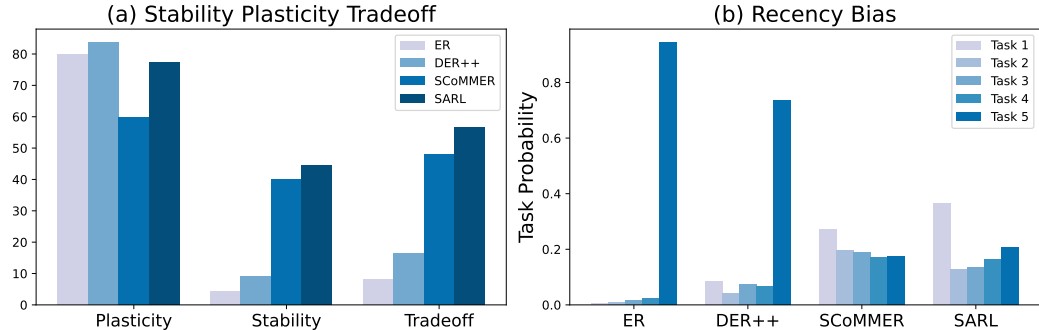

Figure 4: (a) Comparison of the stability and plasticity of the models; and (b) average probability of predicting each task at the end of training for models trained on Seq-CIFAR100 with 200 buffer size. Figure 5 in Appendix shows the task-wise performance of the different models.

structure, allowing the model to incorporate new knowledge without losing the relationships between previously learned classes.

## 5.2 STABILITY-PLASTICITY TRADE-OFF

Continual learning requires balancing plasticity (learning new tasks) and stability (retaining past knowledge). To evaluate this trade-off, we use the metric from (Sarfraz et al., 2022), where plasticity (P) is the average performance on newly learned tasks, and stability (S) is the retention of past tasks after learning task t . The trade-off is given by (2xPxS) / (P + S) . Figure 4(a) shows that SARL significantly enhances stability without sacrificing plasticity, achieving a 362% gain over ER and 127% over DER++. In contrast, SCoMMER, which employs an additional exponentially averaged model for knowledge accumulation, improves stability but severely degrades plasticity over longer task sequences, as reflected in its lower final-task accuracy (47.7% vs. 73.60% for SARL, Figure 5 in Appendix). This highlights the advantage of SARL's semantic-aware sparse learning in fostering a better stability-plasticity balance.

Sequential task learning also introduces a recency bias, favoring recently seen objects (Hou et al., 2019). Figure 4(b) shows that SARL significantly reduces this bias compared to ER and DER++. While SCoMMER's EMA model produces more uniform predictions, it exhibits a declining probability for the last task, which may impact long-term performance. Notably, SARL assigns higher probabilities to earlier tasks, suggesting that its regularization mechanisms effectively preserve the initial function space, mitigating forgetting.

These analyses collectively show that SARL is able to effectively evaluate semantic similarities between objects across tasks and utilize them to enforce a semantic structure in the representation space. This creates a synergy between plasticity and stability, leading to a better balance between the two and reduces forgetting and recency bias.

## 6 CONCLUSION

Inspired by how the brain uses sparse coding and semantic relationships between objects to inform memory formation and retrieval, we proposed semantic-aware representation learning (SARL), which employs sparse activations to create semantically rich object prototypes which can effectively capture the inherent similarities between objects. The semantic structure is leveraged to align new object representations with similar objects from previous tasks while promoting separation from dissimilar ones. Our approach facilitates feature reuse while reducing interfering, creating a synergy between the plasticity and stability of the model. Our empirical evaluation in challenging CL scenarios demonstrates SARL's effectiveness. By enforcing a semantic structure on the representations of objects across tasks, SARL balances model plasticity and stability, reduces task recency bias, and mitigates forgetting. Our findings present a compelling case for incorporating semantic-aware representation learning similar to the brain's method.

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

## A    EXPERIMENTAL SETUP

To ensure fair comparisons between continual learning (CL) methods and isolate algorithmic improvements from training regimen variability, we adopt consistent experimental conditions across all models. In line with Buzzega et al. (2020), we use a ResNet-18 architecture trained with an SGD optimizer and a batch size of 32 for both task data and memory buffer. We use an initial learning rate (lr) of 0.03 with multistep decay scheduler with factor 0.1. Standard data augmentations—random cropping and horizontal flipping—are applied. For all tasks, we include a warm-up stage consisting of 3 epochs, during which the model trains solely on the task-specific data without utilizing memory buffer samples. This allows the model to initialize a strong representation for each task before integrating knowledge from past tasks. For Seq-CIFAR10 we trained for 20 epochs with decay steps For Seq-CIFAR10, the model is trained for 20 epochs with lr decay steps at 15th epoch, for Seq-CIFAR100 and GCIL-CIFAR100, we extend the training to 50 epochs with lr decay steps at 35 and 45. For the Tiny ImageNet dataset, we further increase the number of epochs to 100 with lr decay steps at 70 and 90.

### A.1    SPARSE BACKBONE:

In all experiments, we incorporate activation sparsity by replacing ReLU activations with the k-Winner-Take-All (kWTA) activation function. In kWTA, for each layer, only the top k% of activations at each spatial location are allowed to propagate, while the rest are set to zero. This localized sparsity mechanism ensures that the most relevant features are selected at each spatial location, promoting filter specialization and reducing interference across tasks. The level of sparsity (i.e., k% ) is kept fixed across all layers and experiments. Beyond this change to the activation function, no other modifications are made to the backbone architecture.

### A.2    EVALUATION SETTINGS:

For the Class-IL setting, we follow established baselines and evaluate on three datasets: sequential CIFAR-10 (Seq-CIFAR10) (Krizhevsky et al., 2009), where 10 classes are split into 5 disjoint tasks with 2 classes per task; sequential CIFAR-100 (Seq-CIFAR100) (Krizhevsky et al., 2009), splitting 100 classes into 5 tasks with 20 classes each; and sequential TinyImageNet (Le & Yang, 2015) (Seq-TinyImageNet), where 200 classes are divided into 10 tasks of 20 classes each.

For the Generalized Class-IL (GCIL) setting (Mi et al., 2020), probabilistic modeling is used to randomly vary three task characteristics: the number of classes, the specific classes included, and the sample sizes. As in Sarfraz et al. (2023), we apply this GCIL setting to CIFAR-100, using 20 tasks with 1,000 samples per task and a maximum of 50 classes per task. To separate the effects of class imbalance from the model's ability to learn from recurring classes across tasks, we evaluate the model on both uniform and long-tailed data distributions. For consistency, the GCIL dataset seed is fixed at 1993 for all experiments, ensuring a fair comparison across methods.

For Online-CL, we follow previous works Caccia et al.; Wu et al. (2024) and evaluate our method under the single-head setting. For this setting, we also report average anytime accuracy (**AAA**) Caccia et al., which measures the model's performance throughout the training stream. Let $AA_j$ denote the test average accuracy after training on task $T_j$. Then, the metrics are defined as:

$$AAA = \frac{1}{N} \sum_{j=1}^{N} AA_j, \quad Acc = AA_N,$$

where $N$ is the total number of tasks. Following (Wu et al., 2024), we evaluate our method on the Seq-CIFAR10 and Seq-CIFAR100,. Seq-CIFAR10 comprises 5 tasks, each containing 2 classes, Seq-CIFAR100 consists of 10 tasks with 10 classes each.

## B    ADDITIONAL RESULTS

### B.1    ABLATION STUDY

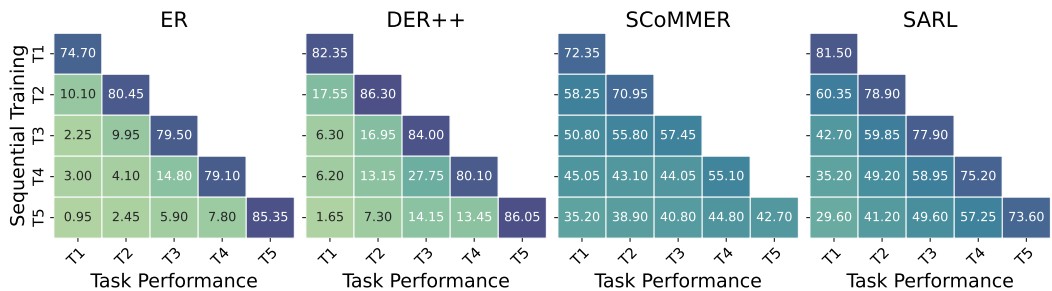

Figure 5: The task-wise performance (x-axis) of the different models assessed after training on each task (y-axis) in Seq-CIFAR100 with a buffer size of 200.

Table 4: Selected hyperparameters for SARL. For all our experiments, we use $\lambda_{SM}$=0.01, $\tau_s$=0.8, and $\beta$=1.

| Dataset | Buffer | kw% | $\lambda_{OP}$ | $\alpha$ |
|---|---|---|---|---|
| Seq-CIFAR10 | 200 | 0.8 | 0.5 | 0.2 |
| | 500 | 0.8 | 0.5 | 0.2 |
| Seq-CIFAR100 | 200 | 0.9 | 0.5 | 0.5 |
| | 500 | 0.9 | 0.5 | 0.2 |
| Seq-TinyImageNet | 200 | 0.9 | 0.2 | 0.2 |
| | 500 | 0.9 | 0.5 | 0.5 |
| GCIL-Uniform | 200 | 0.7 | 0.2 | 0.2 |
| | 500 | 0.7 | 0.2 | 0.2 |
| GCIL-Longtail | 200 | 0.7 | 0.2 | 0.5 |
| | 500 | 0.7 | 0.2 | 0.5 |

Here, we systematically assess the contribution of each key component of SARL by incrementally adding them on top of the baseline ER. Table 5 reveals the importance of each element in improving model performance. Firstly, the inclusion of sparse coding via activation sparsity alone yields a substantial performance boost. This underscores the importance of activation sparsity in mitigating task interference by allowing only the most salient neurons to remain active, effectively controlling task-specific noise and enhancing generalization.

Table 5: Impact of adding distinct elements of SARL on Seq-CIFAR10 performance with 200 buffer size.

| Sparsity | $\mathcal{L}_{OP}$ | $\mathcal{L}_{SM}$ | $\mathcal{L}_{FR}$ | Accuracy |
|---|---|---|---|---|
| ✗ | ✗ | ✗ | ✗ | $44.79_{\pm 1.86}$ |
| ✓ | ✗ | ✗ | ✗ | $58.23_{\pm 1.73}$ |
| ✓ | ✓ | ✗ | ✗ | $62.53_{\pm 0.41}$ |
| ✓ | ✗ | ✓ | ✗ | $61.60_{\pm 0.58}$ |
| ✓ | ✓ | ✓ | ✗ | $63.83_{\pm 0.60}$ |
| ✓ | ✓ | ✓ | ✓ | $70.97_{\pm 0.47}$ |

Next, we observe the separate effects of adding object prototype loss ($\mathcal{L}_{OP}$) and semantic-aware metric learning ($\mathcal{L}_{SM}$) on top of sparsity. Object prototype loss introduces regularization, stabilizing the model by maintaining the average representations of objects, which further improves accuracy to 63.01%. Semantic-aware metric learning, on the other hand, enhances forward transfer by leveraging the semantic similarities between objects while reducing interference. Notably, when both losses are applied together, the synergy between them becomes evident. The object prototype loss ensures stable representations of previously learned object, which in turn helps semantic-aware metric learning enforce a consistent semantic structure throughout training. Finally, adding forward regularization ($\mathcal{L}_{FR}$) to the mix encourages the model to stay within the functional vicinity of previously learned states as new tasks are introduced. This final combination of all components achieves the best performance, highlighting the cohesive benefits of SARL's components working together to reduce interference and improve knowledge retention.

### B.2 RESULTS ON DIFFERENT BACKBONE

To further test the versatility of our approach, we evaluate SARL on the ViT Small backbone and VGG. It is well known that Vision Transformers (ViTs) struggle with smaller datasets due to their inherent architectural requirements, which demand pretraining on large-scale datasets to achieve performance comparable to convolutional networks like ResNet. This limitation is particularly evident in continual learning settings with smaller datasets, leading to a noticeable performance gap between ViT and ResNet backbones.

Despite this inherent challenge, Table 6 shows that SARL provides considerable performance gains over ER across all evaluation settings. These results highlight the robustness and adaptability of SARL, even in scenarios where the backbone architecture is less suited for the dataset size. Additionally, SARL's performance on the VGG backbone is particularly noteworthy. The VGG backbone, which lacks the inductive bias of residual connections and is less efficient compared to modern architectures, still benefits significantly from SARL. Specifically, SARL achieves state-of-the-art results in both Class-IL and Task-IL settings with VGG. This demonstrates SARL's ability to enhance learning effectiveness even in scenarios where the backbone architecture itself has inherent limitations.

Importantly, it should be noted that no hyperparameter tuning was performed specifically for these backbones. Instead, we used the same hyperparameters optimized for the ResNet backbone. This suggests that further improvements in performance are possible by tailoring the hyperparameters to better suit the backbone architecture and adapting the functional regularization. The significant performance improvements observed with SARL validate its capability to enhance learning effectiveness across diverse backbone architectures, emphasizing its potential as a versatile and robust continual learning method.

Table 6: Performance comparison on ViT-Small and VGG backbones. We report the mean and std of three runs.

| Backbone | Buffer | Method | Seq-CIFAR10 | | Seq-CIFAR100 | |
|---|---|---|---|---|---|---|
| | | | Class-IL | Task-IL | Class-IL | Task-IL |
| ViT-Small | 200 | ER | $21.92_{\pm1.70}$ | $69.65_{\pm4.05}$ | $11.90_{\pm1.75}$ | $29.11_{\pm7.05}$ |
| | | SARL | $\mathbf{26.04}_{\pm0.45}$ | $\mathbf{82.74}_{\pm1.39}$ | $\mathbf{19.08}_{\pm3.00}$ | $\mathbf{44.82}_{\pm4.46}$ |
| | 500 | ER | $24.41_{\pm1.46}$ | $72.30_{\pm1.79}$ | $12.51_{\pm1.47}$ | $31.08_{\pm4.25}$ |
| | | SARL | $\mathbf{32.94}_{\pm1.14}$ | $\mathbf{83.25}_{\pm1.86}$ | $\mathbf{22.00}_{\pm0.79}$ | $\mathbf{47.68}_{\pm1.99}$ |
| VGG | 200 | ER | $40.63_{\pm0.74}$ | $80.50_{\pm1.05}$ | $18.71_{\pm0.78}$ | $40.53_{\pm2.85}$ |
| | | SARL | $\mathbf{63.06}_{\pm2.48}$ | $\mathbf{91.07}_{\pm0.84}$ | $\mathbf{26.47}_{\pm0.60}$ | $\mathbf{51.73}_{\pm0.81}$ |
| | 500 | ER | $54.09_{\pm0.87}$ | $86.83_{\pm0.88}$ | $23.01_{\pm0.66}$ | $50.27_{\pm0.91}$ |
| | | SARL | $\mathbf{66.15}_{\pm0.70}$ | $\mathbf{92.37}_{\pm0.42}$ | $\mathbf{33.82}_{\pm0.65}$ | $\mathbf{58.69}_{\pm0.50}$ |

## C ADDITIONAL METRICS

We provide additional metrics for all our experiments to comprehensively evaluate SARL's performance. Since the number of test samples per task is uniform across all tasks, the final accuracy, calculated as the total number of correctly classified samples across all tasks divided by the total number of samples, is equivalent to the average accuracy, which is the mean accuracy across individual tasks at the end of training. Table 7 provides additional metrics such as forgetting, stability, and plasticity, which provide deeper insights into model behavior. Forgetting measures the model's ability to retain knowledge from previous tasks by quantifying the average drop in accuracy of a task at the end of continual learning training compared to its accuracy when first learned, where lower forgetting indicates better knowledge preservation. Stability (S) reflects the average accuracy on previously learned tasks at the end of training, demonstrating how well the model performs on earlier tasks. Plasticity (P), on the other hand, evaluates the model's ability to effectively learn new tasks, measured as the average accuracy of tasks when they are initially trained. Trade-off is measured by (2 * S * P) / (P + S) and measure how well the method keeps a balance between the stability

and plasticity of the model. Together, these metrics offer a holistic view of SARL's performance, capturing its strengths and trade-offs in diverse continual learning scenarios.

Note that the vast majority of baselines do not provide these metrics and hence we cannot compare the baselines across these metrics unfortunately. In addition to the comparison on Average Accuracy in Table 1, we compare the forgetting and additional metrics for the baselines in Table 7 (where available). Note that DER++ is the strongest single model baseline and SARL considerable reduces forgetting by a considerable margin.

Table 7: Performance metrics across datasets and buffer sizes.

| Dataset | Buffer | Accuracy | Forgetting | Stability | Plasticity | Trade-off |
|---------|--------|----------|------------|-----------|------------|-----------|
| Seq-CIFAR10 | 200 | $70.97_{\pm0.47}$ | $14.83_{\pm0.43}$ | $67.72_{\pm1.05}$ | $82.64_{\pm0.73}$ | $74.44_{\pm0.78}$ |
|  | 500 | $75.64_{\pm0.36}$ | $8.75_{\pm6.89}$ | $73.39_{\pm1.00}$ | $85.31_{\pm0.92}$ | $78.90_{\pm0.88}$ |
| Seq-CIFAR100 | 200 | $48.96_{\pm0.53}$ | $33.82_{\pm0.56}$ | $43.12_{\pm0.91}$ | $76.45_{\pm1.01}$ | $55.14_{\pm0.98}$ |
|  | 500 | $55.30_{\pm0.61}$ | $26.78_{\pm1.53}$ | $50.21_{\pm1.39}$ | $76.73_{\pm0.62}$ | $60.69_{\pm0.83}$ |
| Seq-TINYIMG | 200 | $28.95_{\pm1.13}$ | $42.08_{\pm2.94}$ | $23.99_{\pm1.41}$ | $66.82_{\pm1.52}$ | $35.27_{\pm1.29}$ |
|  | 500 | $32.56_{\pm1.23}$ | $27.34_{\pm0.56}$ | $29.88_{\pm1.52}$ | $57.16_{\pm1.73}$ | $39.24_{\pm0.72}$ |

Table 8: Results of different methods on Seq-CIFAR10 and Seq-CIFAR100 datasets with varying buffer sizes.

| Dataset | Buffer | Method | Accuracy | Forgetting | Stability | Plasticity | Trade-off |
|---------|--------|--------|----------|------------|-----------|------------|-----------|
| Seq-CIFAR10 | 200 | ER | $50.36_{\pm2.41}$ | $57.60_{\pm3.53}$ | $38.40_{\pm3.01}$ | $\mathbf{96.44}_{\pm0.42}$ | $54.88_{\pm2.98}$ |
|  |  | DER++ | $65.74_{\pm1.82}$ | $31.75_{\pm2.86}$ | $58.73_{\pm2.53}$ | $91.13_{\pm0.65}$ | $71.40_{\pm1.76}$ |
|  |  | CLSER | $65.92_{\pm0.73}$ | $28.74_{\pm1.45}$ | $67.70_{\pm1.21}$ | $81.17_{\pm1.45}$ | $73.83_{\pm1.21}$ |
|  |  | SCoMMER | $66.80_{\pm0.94}$ | $28.32_{\pm1.35}$ | $62.26_{\pm1.49}$ | $84.72_{\pm1.18}$ | $71.76_{\pm0.78}$ |
|  |  | SARL | $\mathbf{70.97}_{\pm0.47}$ | $\mathbf{14.83}_{\pm0.43}$ | $\mathbf{67.72}_{\pm1.05}$ | $82.64_{\pm0.73}$ | $\mathbf{74.44}_{\pm0.78}$ |
|  | 500 | ER | $62.70_{\pm1.04}$ | $41.92_{\pm1.98}$ | $53.92_{\pm1.34}$ | $\mathbf{96.24}_{\pm0.86}$ | $69.11_{\pm1.04}$ |
|  |  | DER++ | $70.10_{\pm1.47}$ | $26.79_{\pm1.08}$ | $63.70_{\pm1.89}$ | $91.53_{\pm0.64}$ | $75.11_{\pm1.51}$ |
|  |  | CLSER | $75.16_{\pm0.86}$ | $22.56_{\pm1.04}$ | $\mathbf{73.40}_{\pm0.33}$ | $78.03_{\pm0.81}$ | $75.64_{\pm0.57}$ |
|  |  | SCoMMER | $74.25_{\pm0.42}$ | $20.13_{\pm1.30}$ | $70.62_{\pm0.45}$ | $87.99_{\pm0.45}$ | $78.36_{\pm0.37}$ |
|  |  | SARL | $\mathbf{75.64}_{\pm0.36}$ | $\mathbf{8.75}_{\pm6.89}$ | $73.39_{\pm1.00}$ | $85.31_{\pm0.92}$ | $\mathbf{78.90}_{\pm0.88}$ |
| Seq-CIFAR100 | 200 | ER | $21.73_{\pm0.03}$ | $76.13_{\pm0.29}$ | $5.29_{\pm0.15}$ | $\mathbf{82.63}_{\pm0.21}$ | $9.94_{\pm0.26}$ |
|  |  | DER++ | $30.68_{\pm1.35}$ | $65.10_{\pm1.74}$ | $17.19_{\pm1.75}$ | $82.77_{\pm0.72}$ | $28.44_{\pm2.37}$ |
|  |  | CLSER | $43.80_{\pm1.89}$ | $36.33_{\pm1.46}$ | $\mathbf{45.75}_{\pm1.78}$ | $50.47_{\pm1.03}$ | $47.99_{\pm1.35}$ |
|  |  | SCoMMER | $40.74_{\pm0.53}$ | $42.20_{\pm1.13}$ | $35.98_{\pm0.90}$ | $63.30_{\pm6.21}$ | $45.80_{\pm1.76}$ |
|  |  | SARL | $\mathbf{48.96}_{\pm0.53}$ | $33.82_{\pm0.56}$ | $43.12_{\pm0.91}$ | $76.45_{\pm1.01}$ | $\mathbf{55.14}_{\pm0.98}$ |
|  | 500 | ER | $28.03_{\pm1.06}$ | $67.08_{\pm1.11}$ | $13.42_{\pm1.60}$ | $81.69_{\pm0.17}$ | $23.03_{\pm2.37}$ |
|  |  | DER++ | $42.03_{\pm2.01}$ | $48.90_{\pm2.42}$ | $32.01_{\pm2.74}$ | $81.15_{\pm0.80}$ | $45.87_{\pm2.89}$ |
|  |  | CLSER | $51.68_{\pm0.99}$ | $30.74_{\pm0.25}$ | $\mathbf{52.18}_{\pm0.27}$ | $58.07_{\pm2.17}$ | $54.95_{\pm1.04}$ |
|  |  | SCoMMER | $50.11_{\pm0.31}$ | $32.98_{\pm0.80}$ | $46.40_{\pm0.79}$ | $66.93_{\pm1.72}$ | $54.79_{\pm0.35}$ |
|  |  | SARL | $\mathbf{55.30}_{\pm0.61}$ | $\mathbf{26.78}_{\pm1.53}$ | $50.21_{\pm1.39}$ | $\mathbf{76.73}_{\pm0.62}$ | $\mathbf{60.69}_{\pm0.83}$ |

## D   HYPERPARAMETER TUNING

During hyperparameter tuning, we used a small validation set to adjust values for kw%, $\lambda_{OP}$, and the coefficient $\alpha$ for $\mathcal{L}_{FR}$. For all experiments, we set $\tau_s = 0.8$, $\beta = 1$ and $\lambda_{SM} = 0.01$. The value of kw% was selected from the set $\{0.7, 0.8, 0.9\}$, while $\lambda_{OP}$ and $\alpha$, were chosen from $\{0.2, 0.5, 1\}$. The final selected parameters are presented in Table 4. It is important to note that we did not perform an exhaustive hyperparameter search, so better configurations may exist. Furthermore, we observed a synergistic relationship among the parameters, and the model demonstrated robustness to a range of values, which significantly eased the tuning process.

To further illustrate this, the results in Table 9 demonstrate that SARL is not overly sensitive to specific hyperparameter values, as multiple configurations yield consistent performance improvements. For instance, across different values of $\alpha$, $\lambda_{OP}$, and kw%, the method achieves robust

Class-IL and Task-IL accuracies on Seq-CIFAR10, with minor variations in performance. While certain configurations (e.g., $\alpha = 0.2$, $\lambda_{OP} = 0.5$, kw% = 0.8 ) achieve the highest Class-IL accuracy (70.97%), other values (e.g., kw% = 0.9, $\alpha = 0.2$, $\lambda_{OP} = 0.5$ ) provide similar results (70.08%). We present results for all the grid values explored during hyperparameter tuning, demonstrating that SARL achieves consistent performance improvements across various configurations without relying heavily on precise hyperparameter adjustments. This reliability underscores SARL's practicality and suitability for a wide range of continual learning applications

Table 9: Sensitivity to hyperparameters on the performance of SARL. For all the experiments, we use learning rate $\eta = 0.1$, $\alpha = 0.999$, $\lambda_{OP} = 0.5$, $\lambda_{SA} = 0.01$, $\lambda_{CR} = 0.15$ and $\tau = 0.8$.

| | | | seq-cifar10 | |
|---|---|---|---|---|
| kw% | $\alpha$ | $\lambda_{OP}$ | Class-IL | Task-IL |
| | | 0.2 | $70.73_{\pm 2.31}$ | $95.77_{\pm 0.16}$ |
| | 0.2 | 0.5 | $70.97_{\pm 0.47}$ | $95.72_{\pm 0.36}$ |
| | | 1 | $70.57_{\pm 0.73}$ | $95.71_{\pm 0.08}$ |
| | | 0.2 | $68.53_{\pm 1.14}$ | $95.23_{\pm 0.07}$ |
| 0.8 | 0.5 | 0.5 | $69.40_{\pm 1.21}$ | $95.21_{\pm 0.19}$ |
| | | 1 | $68.79_{\pm 0.84}$ | $95.14_{\pm 0.09}$ |
| | | 0.2 | $65.30_{\pm 1.40}$ | $93.73_{\pm 0.68}$ |
| | 1 | 0.5 | $64.59_{\pm 0.62}$ | $93.72_{\pm 0.42}$ |
| | | 1 | $63.86_{\pm 1.45}$ | $93.19_{\pm 0.08}$ |
| | | 0.2 | $70.05_{\pm 1.44}$ | $95.49_{\pm 0.14}$ |
| | 0.2 | 0.5 | $70.08_{\pm 1.48}$ | $95.63_{\pm 0.22}$ |
| | | 1 | $69.07_{\pm 0.68}$ | $95.52_{\pm 0.01}$ |
| | | 0.2 | $68.12_{\pm 2.90}$ | $95.13_{\pm 0.41}$ |
| 0.9 | 0.5 | 0.5 | $66.66_{\pm 0.55}$ | $94.92_{\pm 0.31}$ |
| | | 1 | $65.49_{\pm 0.08}$ | $94.48_{\pm 0.38}$ |
| | | 0.2 | $63.99_{\pm 2.83}$ | $93.34_{\pm 0.91}$ |
| | 1 | 0.5 | $64.04_{\pm 0.06}$ | $93.00_{\pm 1.05}$ |
| | | 1 | $63.80_{\pm 1.56}$ | $92.87_{\pm 1.33}$ |

Table 10: Memory and performance comparison of SARL and baselines on Seq-CIFAR100. #M stands for number of models, #FWD for number of forward passes, and $|B|$ for buffer size.

| Method | #M | #FWD (Tasks) | #FWD (Buffer) | $|B|$ (**MB**) | Total Memory (MB) | Memory Ratio (vs ER) | Avg. Acc (%) | Avg. Acc / Memory Ratio |
|---|---|---|---|---|---|---|---|---|
| ER | 1 | 1 | 1 | 5.86 | 48.52 | 1.0000 | 28.02 | 28.02 |
| DER++ | 1 | 1 | 2 | 6.05 | 48.71 | 1.0039 | 41.40 | 41.24 |
| CLS-ER | 3 | 1 | 3 | 5.86 | 133.84 | 2.7584 | 51.40 | 18.63 |
| SCoMMER | 2 | 1 | 2 | 5.86 | 91.18 | 1.8792 | 49.63 | 26.41 |
| SARL | 1 | 1 | 1 | 8.15 | 50.09 | 1.0323 | **55.30** | **53.57** |

# E   SEMANTIC SIMILARITY BETWEEN CLUSTERS

To further strengthen our analysis in Figure 2 which provides similarity matrices for dense and sparse activations, we defined two clusters within the CIFAR-10 classes: the Animal Cluster, comprising 'bird,' 'cat,' 'deer,' 'dog,' 'frog,' and 'horse,' and the Vehicle Cluster, consisting of 'airplane,' 'automobile,' 'ship,' and 'truck.' The comparison of inter- and intra-cluster similarities is visualized in Figure 6, and the results indicate that sparse activations exhibit consistently higher average similarity within clusters compared to dense activations. For the Animal Cluster, sparse activations achieve an average similarity of 0.8873 compared to 0.8367 for dense activations. Similarly, for the Vehicle Cluster, sparse activations achieve 0.8694 compared to 0.8684 for dense activations. This suggests that sparse activations better group objects within the same semantic category, capturing seman-

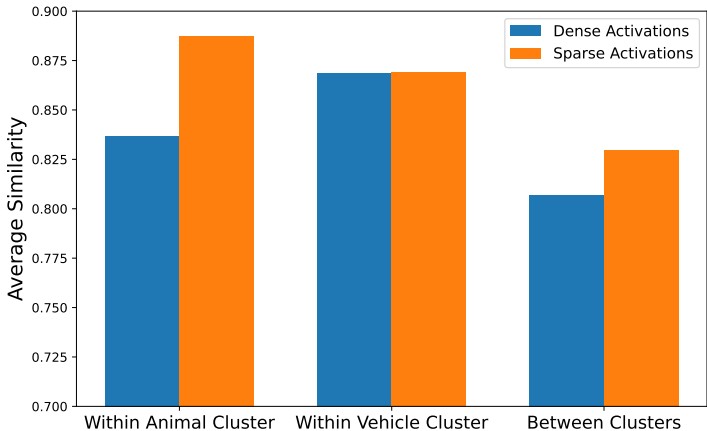

Figure 6: Inter- and Intra-Cluster Similarities Across Animal and Vehicle Clusters in CIFAR-10 for Dense and Sparse Activations

tic cohesion more effectively. By activating only the most relevant neurons, sparse representations reduce noise and focus on features critical for semantic grouping.

Sparse activations also demonstrate a slightly higher average similarity between clusters, with 0.8295 compared to 0.8068 for dense activations. However, this inter-cluster similarity is significantly lower than the intra-cluster similarity, aligning with the expected semantic distinction between clusters. Notably, sparse representations encode inter-cluster relationships more semantically rather than merely enforcing hard separations. For instance, airplane and bird, which share abstract features like wings and the ability to fly, exhibit a higher similarity of 0.90 with sparse activations compared to 0.87 with dense activations. In contrast, ship and horse, which are semantically dissimilar, show a similarity of 0.81 with sparse activations compared to 0.83 with dense activations. These findings reinforce our hypothesis that sparse activations can enhance the model's capability to capture semantic structure across objects.

## F VISUALIZATION OF REPRESENTATION SPACE

To further analyze the representations and evaluate how well SARL captures and enforces a semantic structure among objects, we project the object prototypes into a two-dimensional space using t-SNE and compare SARL with ER on Seq-CIFAR10 with a buffer size of 200 (Figure 7). The t-SNE visualizations reveal clear differences in the representational quality of SARL compared to ER. In the left panel (ER), object prototypes exhibit a lack of structured clustering, with semantically similar classes, such as animals ("bird," "frog," "deer"), being dispersed and overlapping with other categories. This indicates that ER struggles to effectively group related objects or separate distinct semantic categories in the representation space. In contrast, the right panel (SARL) demonstrates significant improvements, forming compact and cohesive clusters for semantically similar objects. For example, animals are tightly grouped together, and SARL effectively captures the semantic similarity between "bird" and "airplane," a relationship that ER fails to encode. This structured organization of prototypes highlights SARL's ability to leverage semantic relationships to guide representation learning, aligning similar classes while maintaining clear distinctions between unrelated ones. These findings underscore the effectiveness of SARL in creating a more meaningful, semantically aware, and structured representation space.

## G EXTENDED RELATED WORKS

Continual learning has seen significant advancements in recent years, with diverse approaches targeting the challenges of catastrophic forgetting, model stability, and effective knowledge transfer across tasks. In this section, we extend our related works section and position our SARL in the broader context of these developments.

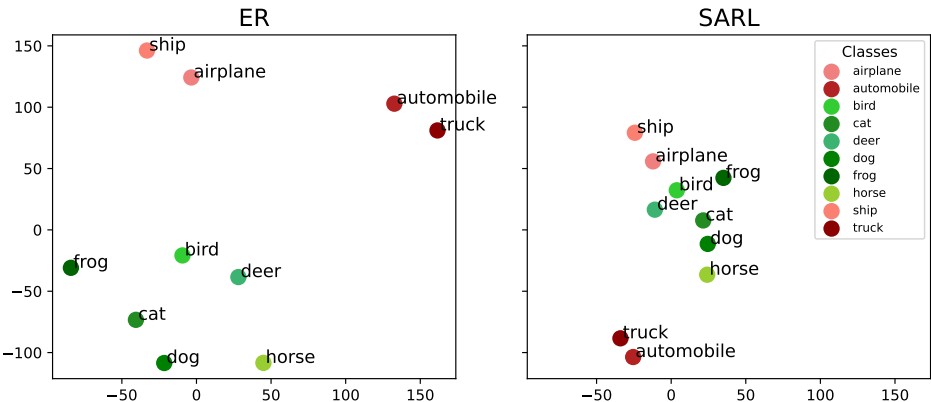

Figure 7: t-SNE plot of object prototypes using trained model with ER and SARL on Seq-CIFAR10 with 200 buffer size.

Meta-learning approaches have shown promise in continual learning by leveraging task-specific adaptations. VR-MCL (Wu et al., 2024) proposes an online CL approach that combines Meta-CL with regularization to efficiently approximating online Hessians, enhancing adaptability in real-time learning scenarios. While our work adopts a batch training setup, allowing multiple passes over sequential task data which we believe is more aligned with many real-world scenarios where datasets can be revisited periodically (e.g., staged data collection or iterative training), VR-MCL provides a focuses on online single-pass learning. The fundamental differences in training settings make direct comparisons challenging, but we recognize the potential of such meta-learning techniques to inspire future extensions of SARL in online CL. Interactive Continual Learning (ICL) (Qi et al., 2024) utilizes multimodal large language models (LLMs) to achieve state-of-the-art performance in continual learning. By leveraging extensive pretraining on diverse datasets (often a superset of classes in the baseline CL datasets) and significantly larger architectures, this approach gains a lot of advantage over the standard setting. It also makes them infeasible for many CL applications due to the magnitudes larger memory and computational costs. The reliance on pretraining introduces an inherent disparity when comparing with SARL, which operates with a randomly initialized model and focuses on both learning robust representations in the CL regime and retaining them as training progresses. Furthermore, Memory-based approaches are widely used in rehearsal-based CL to alleviate catastrophic forgetting. Bilateral Memory Consolidation (BiMeCo) (Nie et al., 2023) introduces a method that consolidates long-term and short-term memories to improve performance. Their approach is complementary to SARL and could potentially enhance SARL further if integrated. However, their evaluation setting of involves pretraining on a large number of base classes before sequential learning, whereas SARL and its baselines initiate continual learning with a randomly initialized model. This distinction in experimental setups underscores the importance of careful contextualization when comparing results across studies.

Prompt-based methods, such as L2P (Wang et al., 2022c) and DualPrompt (Wang et al., 2022b), utilize pre-trained backbones, like those trained on ImageNet, to achieve strong performance in class incremental learning setting. These pre-trained backbones inherently include representations for many classes encountered later in CL tasks or for classes that are semantically similar. This allows these methods to bypass one of the core challenges of CL: learning and preserving general representations of objects across sequential tasks. In contrast, SARL operates in a more challenging setting, starting from a randomly initialized state without leveraging pretraining. Extending SARL to integrate pre-trained models could be a promising direction for future research, but the current work focuses on enabling robust representation learning from a random start utilizing the semantic structure to learn and retain representations.

We would like to emphasize that evaluating CL methods across diverse baselines presents several challenges. Variations in architectures, the use of additional pretraining, access to implicit or explicit auxiliary information, and differences in task structures and metrics significantly influence reported results. Uniform evaluation settings are critical for meaningful assessments, as discrepancies in setup may reflect differences in methodology rather than genuine improvements.

