# OpenReview forum: "Semantic Aware Representation Learning for Lifelong Learning"
_ICLR.cc/2025/Conference — ICLR 2025 Poster_

### Official Review · Reviewer_6oW7 · 2024-10-31

**Soundness:** 3
**Presentation:** 3
**Contribution:** 2
**Rating:** 6
**Confidence:** 2

**Summary:**

This paper introduces Semantic-Aware Representation Learning (SARL), a method aimed at mitigating catastrophic forgetting in lifelong learning. Inspired by human cognitive processes, SARL uses sparse activation and object prototypes to capture semantic relationships, employing semantic-aware metric learning to cluster similar objects while separating dissimilar ones. Experimental results show that SARL excels in Class-Incremental and Generalized Class-Incremental Learning, effectively retaining and transferring knowledge within a single-model setup. SARL’s integration of semantic structure and sparse representations offers a balanced approach to model stability and adaptability in lifelong learning.

**Strengths:**

The SARL method introduces an innovative approach by drawing inspiration from biological processes to incorporate semantic relationships into representation learning.

By employing semantic-aware metric learning, SARL effectively clusters similar objects while separating dissimilar ones, achieving a valuable balance between knowledge retention and adaptation to new information.

Validated across complex incremental learning tasks, such as Class-Incremental and Generalized Class-Incremental Learning, SARL demonstrates robust performance, minimizing task interference and enabling seamless knowledge transfer, highlighting its practical effectiveness.

The paper is well-written, with a clear structure that is easy to follow.

**Weaknesses:**

The objective function (7) of the proposed SARL method includes five loss terms, but the paper lacks a sensitivity analysis on the weights hyperparameter assigned to each term. Such an analysis would provide valuable insights into the impact of each loss term on the model’s performance. The current one in Appendix is not sufficient.

The introduction of object prototypes and semantic-aware metric learning may increase the model’s computational overhead and resource requirements compared to baseline method such as ER. The paper does not provide a detailed discussion on the model’s efficiency

The experiments rely solely on the ResNet-18 backbone, lacking comparisons with other backbone models such as ViT, which could provide a broader evaluation of the method’s effectiveness.

**Questions:**

In the ablation study presented in Table 3, why is the performance in the first row so low? Is this row representing a traditional ResNet-18 classifier?

Do you have any insights into why the improvements for the Seq-CIFAR10 dataset are marginal compared to the more significant gains observed with Seq-CIFAR100 and Seq-TinyImageNet?

---

> ### Author Response · Authors · 2024-11-23
> **Author's response (1/2)**
>
> Thank you for recognizing the novelty of SARL’s biologically inspired approach and its practical effectiveness in reducing task interference. We also appreciate your positive remarks on the clarity and structure of the paper. Your comments have provided us with valuable directions for further strengthening our work, and we address each of your points below in detail.
>
> > The objective function (7) of the proposed SARL method includes five loss terms, but the paper lacks a sensitivity analysis on the weights hyperparameter assigned to each term. Such an analysis would provide valuable insights into the impact of each loss term on the model’s performance.
>
> Thank for for the valuable suggestion. We have added hyper paramter sensitivity to the Appendix of the revised manuscript. Table 7 demonstrate that SARL is not overly sensitive to specific hyperparameter values, as multiple configurations yield consistent performance improvements. For instance, across different values of  $\alpha$ ,  $\lambda_{OP}$ , and  kw}% , the method achieves robust Class-IL and Task-IL accuracies on Seq-CIFAR10, with minor variations in performance. While certain configurations (e.g.,  $\alpha$ = 0.2, $\lambda_{OP} = 0.5$, kw\% = 0.8 ) achieve the highest Class-IL accuracy (70.97\%), other values (e.g.,  \kw% = 0.9, $\alpha$ = 0.2, $\lambda_{OP}$ = 0.5 ) provide similar results (70.08\%). We present results for all the grid values explored during hyperparameter tuning, demonstrating that SARL achieves consistent performance improvements across various configurations without relying heavily on precise hyperparameter adjustments. This reliability underscores SARL’s practicality and suitability for a wide range of continual learning applications.
>
> > The introduction of object prototypes and semantic-aware metric learning may increase the model’s computational overhead and resource requirements compared to baseline method such as ER. The paper does not provide a detailed discussion on the model’s efficiency
>
> Thank you for the valuable suggestion, we added a discussion on memory requirements in the reopo. SARL maintains an episodic memory for replay (images and logits) and a compact set of object prototypes for efficient retrieval and learning. Additionally, at the beginning of each task, we store the logits for the current task samples to enable functional regularization. Below, we provide a detailed breakdown of the memory requirements for SARL, quantifying each component’s footprint.
>
> For all components, data is stored as 32-bit floating-point values:
> - Memory for 500 images in the buffer: 500 \times 3 \times 32 \times 32 \times 4 bytes = 5.86 MB.
> - Memory for logits of 500 images in the buffer: 500 \times 100 \times 4 bytes = 0.19 MB.
> - Memory for logits at the beginning of the task (functional regularization): 10 \times 500 \times 100 \times 4 bytes = 1.9 MB.
> - Memory for object prototypes (100 classes, 512-dimensional features): 100 \times 512 \times 4 bytes = 0.20 MB.
> - Model size (ResNet-18): 42.66 MB.
>
> As summarized in the table below, the additional memory overhead introduced by SARL compared to Baseline ER is negligible, while providing substantial performance improvements. Furthermore, SARL is significantly more memory-efficient than strong baselines like SCoMMER and CLS-ER, which utilize multiple models and forward passes.
>
> **Memory and Performance Comparison on Seq-CIFAR100 (500 Buffer Size):**
> | Method   | # Models | # FWD Passes over Task Samples | # FWD Passes over Memory Samples | Memory Buffer Size (MB) | Total Memory (MB) | Memory Ratio (vs ER) | Avg. Acc (%) | Avg. Acc / Memory Ratio |
> |----------|----------|--------------------------------|-----------------------------------|-------------------------|-------------------|-----------------------|--------------|--------------------------|
> | ER       | 1        | 1                              | 1                                 | 5.86                   | 48.52             | 1.000                 | 28.02        | 28.02                   |
> | DER++    | 1        | 1                              | 2                                 | 5.86 + 0.19            | 48.71             | 1.0039                | 41.4         | 41.24                   |
> | CLS-ER   | 3        | 1                              | 3                                 | 5.86                   | 133.84            | 2.7584                | 51.4         | 18.63                   |
> | SCoMMER  | 2        | 1                              | 2                                 | 5.86                   | 91.18             | 1.8792                | 49.63        | 26.41                   |
> | SARL     | 1        | 1                              | 1                                 | 5.86 + 0.19 + 1.9 + 0.20 | 50.09             | 1.0323                | **55.3**     | **53.57**               |

---

> ### Author Response · Authors · 2024-11-23
> **Author's response (2/2)**
>
> > The experiments rely solely on the ResNet-18 backbone, lacking comparisons with other backbone models such as ViT, which could provide a broader evaluation of the method’s effectiveness.
>
> Thank you for this suggestion. We focused on ResNet-18 in the current experiments to maintain consistency with previous works in continual learning, but we agree that evaluating SARL on additional architectures would be valuable. we are actively running experiments on ViT-Small without pretraining to further evaluate SARL’s effectiveness on modern backbone architectures and will update the results here. We would like to highlight that SARL is independent of the choice of backbone and its principles are broadly applicable across different architectures.
>
> > In the ablation study presented in Table 3, why is the performance in the first row so low? Is this row representing a traditional ResNet-18 classifier?
>
> In the ablation study presented in Table 3, the first row represents the baseline ER (Experience Replay) method with a ResNet-18 backbone, using only cross-entropy loss on the buffered samples, without any additional knowledge retention or regularization mechanisms. This setup serves as the starting point for evaluating the impact of the components introduced in SARL. Since the buffer size is small, ER struggles to effectively retain information from previous tasks, resulting in significant forgetting and lower overall performance. This highlights the limitations of relying solely on a simple replay mechanism in scenarios with constrained memory resources, emphasizing the importance of the semantic-aware representation learning and prototype-based mechanisms introduced in SARL. By addressing these limitations, SARL demonstrates substantial performance improvements over this baseline.
>
> > Do you have any insights into why the improvements for the Seq-CIFAR10 dataset are marginal compared to the more significant gains observed with Seq-CIFAR100 and Seq-TinyImageNet?
>
> We believe that the smaller gains on Seq-CIFAR10 likely stem from the lower task complexity and fewer classes in Seq-CIFAR10 compared to Seq-CIFAR100 and Seq-TinyImageNet. Since Seq-CIFAR10 has only five tasks, each with just two classes, the opportunity for catastrophic forgetting is naturally reduced, and semantic relationships between classes are less complex. In contrast, Seq-CIFAR100 and Seq-TinyImageNet involve more classes with greater semantic overlap, which enhances SARL’s benefit in utilizing semantic relationships to minimize forgetting and improve feature reuse.
>
> We hope these responses address your comments and concerns thoroughly. We are grateful for your feedback, which has significantly helped us enhance the clarity and depth of our work.

---

> > ### Comment · Reviewer_6oW7 · 2024-11-23
> > **Feedback on Author(s)’ Response**
> >
> > Dear Author(s),
> >
> > I appreciate your efforts in adding experiments on sensitivity analysis and memory efficiency, as these results appear promising. Your explanations addressing the questions I raised about certain experimental results are reasonable and convincing. While I acknowledge that the proposed framework is model-agnostic, I believe additional experiments using other backbone models would further validate its effectiveness. I would be happy to adjust my score accordingly after such additions.
> >
> > Yours,

---

> > > ### Author Response · Authors · 2024-11-29
> > > **Author's response to Reviewer's Feedback**
> > >
> > > Thank you for your encouraging feedback. We sincerely appreciate your acknowledgment of the sensitivity analysis and memory efficiency experiments, as well as your thoughtful suggestions for further validating SARL’s effectiveness. Your input has been instrumental in improving our work. We completed our experiments on the different backbones.
> > >
> > > To further validate the versatility of SARL, we extended our experiments to include two additional backbones: ViT-Small and VGG. These are included in the Appendix Section B.2. Below, we summarize the key findings:
> > >
> > > **ViT-Small Backbone**:
> > >
> > > Vision Transformers (ViTs), such as ViT-Small, are known to struggle with smaller datasets due to their architectural design, which demands pretraining on large-scale datasets for optimal performance. In consistency with our goal of learning robust representations in continual learning settings, we opted for training ViT-Small from scratch. Despite these challenges, SARL demonstrates considerable performance gains over the ER baseline across all evaluation settings. This underscores SARL’s robustness and adaptability, even when the backbone architecture is less suited for the dataset size.
> > >
> > > **VGG Backbone**:
> > >
> > > The VGG backbone, which lacks modern architectural enhancements like residual connections, poses unique challenges. Following many recent works, we used the VGG-16 version with batch normalization. SARL achieves considerable performance improvements over ER.
> > >
> > >
> > > #### Table: Performance comparison on ViT-Small and VGG backbones. We report the mean and std of three runs.
> > >
> > > | **Backbone** | **Buffer** | **Method** | **Seq-CIFAR10** Class-IL | **Seq-CIFAR10** Task-IL | **Seq-CIFAR100** Class-IL | **Seq-CIFAR100** Task-IL |
> > > |--------------|------------|------------|--------------------------|--------------------------|---------------------------|---------------------------|
> > > | **ViT-Small** | 200       | ER         | 21.92 ± 1.70            | 69.65 ± 4.05            | 11.90 ± 1.75             | 29.11 ± 7.05             |
> > > |              |            | SARL       | **26.04 ± 0.45**        | **82.74 ± 1.39**        | **19.08 ± 3.00**         | **44.82 ± 4.46**         |
> > > |              | 500        | ER         | 24.41 ± 1.46            | 72.30 ± 1.79            | 12.51 ± 1.47             | 31.08 ± 4.25             |
> > > |              |            | SARL       | **32.94 ± 1.14**        | **83.25 ± 1.86**        | **22.00 ± 0.79**         | **47.68 ± 1.99**         |
> > > | **VGG**      | 200       | ER         | 40.63 ± 0.74            | 80.50 ± 1.05            | 18.71 ± 0.78             | 40.53 ± 2.85             |
> > > |              |            | SARL       | **63.06 ± 2.48**        | **91.07 ± 0.84**        | **26.47 ± 0.60**         | **51.73 ± 0.81**         |
> > > |              | 500        | ER         | 54.09 ± 0.87            | 86.83 ± 0.88            | 23.01 ± 0.66             | 50.27 ± 0.91             |
> > > |              |            | SARL       | **66.15 ± 0.70**        | **92.37 ± 0.42**        | **33.82 ± 0.65**         | **58.69 ± 0.50**         |
> > >
> > >
> > > Notably, we used the same hyperparameters optimized for the ResNet backbone without any additional tuning for these experiments. This demonstrates SARL’s robustness to hyperparameter changes and suggests that further performance improvements may be achievable by tailoring the hyperparameters to better suit the specific backbone architecture.
> > >
> > > These results validate SARL’s capability to deliver significant performance improvements across diverse backbone architectures, highlighting its potential as a versatile and robust continual learning method.
> > >
> > > We are grateful for your suggestion to include experiments with different backbones, as it allowed us to further explore and validate the generalizability of SARL. We hope these results address your concerns and demonstrate SARL’s effectiveness in varying scenarios.
> > >
> > > Please also see our new results in the Online Continual learning (Online CL) setting (Section B.1 and Summarized in General comments).Online CL is a challenging setting where models process each data instance only once, requiring dynamic learning without revisiting previous data. While SARL is not explicitly designed for Online CL, we adapted it to this setting by dynamically maintaining feature sums and sample counts to compute object prototypes. Despite this adaptation, SARL demonstrates strong performance across datasets, providing gains over recent state of the art which are specifically designed for this setting.
> > >
> > > We are encouraged by the promising results SARL achieved in the Online CL setting and its versatility across diverse datasets. While we believe these results demonstrate SARL’s robustness and adaptability, we would greatly appreciate any suggestions you might have on how we could further improve or enhance your confidence in our work. Your insights have been invaluable, and we welcome any additional feedback to strengthen the manuscript.

---

> > > > ### Comment · Area_Chair_iJoP · 2024-11-30
> > > >
> > > > Dear Reviewer,
> > > >
> > > > The authors have provided their responses. Could you please review them and share your feedback?
> > > >
> > > > Thank you!

---

> ### Comment · Reviewer_6oW7 · 2024-12-02
> **Feedback on Author(s)’ Response**
>
> Thank you for adding those experiments, which make the conclusion more robust. I have no further questions at this time and have raised my overall score from 5 to 6.

---

> > ### Author Response · Authors · 2024-12-02
> > **Author's response to reviewer's comment**
> >
> > Thank you for your kind words and for acknowledging the additional experiments we conducted. We are grateful for your thoughtful feedback throughout the review process, which has helped strengthen our paper. We also sincerely appreciate you raising your overall score, as it reflects your confidence in the improvements we have made.
> >
> > As the rebuttal discussion period is coming to an end, we would like to ask if there is any additional information or analysis we can provide to further increase your confidence in our work. Your insights have been invaluable, and we would be happy to address any remaining questions or suggestions you might have.
> >
> > Thank you once again for your time and effort in reviewing our submission.

---

### Official Review · Reviewer_UqLc · 2024-11-01

**Soundness:** 3
**Presentation:** 2
**Contribution:** 2
**Rating:** 3
**Confidence:** 5

**Summary:**

The paper proposes a method to reduce catastrophic forgetting in continual learning by taking advantage of several aspects: 1) the semantic relationships between learned classes, 2) representation sparsity, and 3) prototype regularization and other losses. The method draws inspiration from the brain’s sparse coding approach and employs activation sparsity to facilitate semantic information encoding in representations to promote reusability and reduce forgetting.

**Strengths:**

1. The paper is clear and easy to read.
2. The combination of losses is interesting.
3. The performance results are very good.

**Weaknesses:**

1. The introduction could benefit from being more specific and quicker in presenting the problem, current limitations, and how the paper addresses these issues in the technical section. A more concise and more streamlined rewrite is recommended.

2. The novelty appears to be somewhat limited, as deep neural networks inherently build and use semantic representations. A clearer motivation for the novel use of Eqs. (1), (2), ..., (5) (6) would be beneficial. Additionally, an introduction to these losses/similarities should be provided to ensure the reviewer understands they are not being presented as novel contributions. Clarification is needed on how the proposed approach to semantic representations differs from or improves upon standard deep learning methods. Additionally, the motivation and novelty behind the specific formulations in Eqs. (1)-(6) should be explained, and it should be explicitly stated which components are novel contributions versus established techniques.

3. In the reviewer's opinion, the paper is a solid combination of well-established techniques that achieves good performance. However, it may be worth considering whether a different venue might be a better fit for this type of contribution. ICLR often prioritizes papers with some analytical theoretical analysis, while contributions that mainly involve combining established techniques with primarily empirical evidence may be less prioritized.

4. The content on lines 054 to 055 is somewhat generic and does not sufficiently characterize the current limitations and issues. A more detailed description, including a toy problem to demonstrate qualitatively how this affects catastrophic forgetting in practice, would be beneficial. The reviewer may not fully grasp all the details of the paper and the insights that led to the proposed approach; therefore, proposing a concrete toy problem may not be straightforward, as it typically depends on the specific effect intended to be demonstrated. One possible suggestion is to choose a small dataset, such as CIFAR-10, and represent it in a 2D or 3D feature space, or, if >3D, using a scatterplot of the feature distribution or a dimensionality reduction technique like t-SNE to illustrate how the semantic aspects contribute as proposed by the authors in the introduction. These papers present some classic methods for visualizing feature representations:

- Wen, Yandong, et al. "A discriminative feature learning approach for deep face recognition." Computer vision–ECCV 2016: 14th European conference, amsterdam, the netherlands, October 11–14, 2016, proceedings, part VII 14. Springer International Publishing, 2016.

- Pernici, Federico, et al. "Maximally compact and separated features with regular polytope networks." The IEEE Conference on Computer Vision and Pattern Recognition (CVPR) Workshops. IEEE, 2019.

**Questions:**

Please refer to the weaknesses section.

---

> ### Author Response · Authors · 2024-11-22
> **Author's Response (1/2)**
>
> Thank you for your valuable feedback. We appreciate your recognition of the clarity of the paper and the strong empirical  results.  We address each of your points in detail below.
>
> > The introduction could benefit from being more specific and quicker in presenting the problem, current limitations, and how the paper addresses these issues in the technical section. A more concise and more streamlined rewrite is recommended.
>
> Thank you for this suggestion. We understand that a clearer, more focused introduction can better frame the specific challenges and contributions of our approach. We have made an attempt to make the introduction concise and highlight the limitations and challenges of existing approaches more clearly. We would really appreciate it if you could provide more specific examples of what you found as redundant in the current introduction and we would be happy to revise accordingly.
>
> > The content on lines 054 to 055 is somewhat generic and does not sufficiently characterize the current limitations and issues. A more detailed description, including a toy problem to demonstrate qualitatively how this affects catastrophic forgetting in practice, would be beneficial. One possible suggestion is to choose a small dataset, such as CIFAR-10, and represent it in a 2D or 3D feature space ...
>
> Thank you for the valuable suggestion to use a more illustrative example and visualizing the representations using t-SNE. We have added two sections in Appendix related to them.
>
> - **t-SNE visualization of the representation space:**
> To analyze the representations and evaluate how well SARL captures and enforces a semantic structure among objects, we project the object prototypes into a two-dimensional space using t-SNE and compare SARL with ER on Seq-CIFAR10 with a buffer size of 200 (Figure 7 in Appendix). The t-SNE visualizations reveal clear differences in the representational quality of SARL compared to ER. In the left panel (ER), object prototypes exhibit a lack of structured clustering, with semantically similar classes, such as animals (“bird,” “frog,” “deer”), being dispersed and overlapping with other categories. This indicates that ER struggles to effectively group related objects or separate distinct semantic categories in the representation space. In contrast, the right panel (SARL) demonstrates significant improvements, forming compact and cohesive clusters for semantically similar objects. For example, animals are tightly grouped together, and SARL effectively captures the semantic similarity between “bird” and “airplane,” a relationship that ER fails to encode. This structured organization of prototypes highlights SARL’s ability to leverage semantic relationships to guide representation learning, aligning similar classes while maintaining clear distinctions between unrelated ones. These findings underscore the effectiveness of SARL in creating a more meaningful, semantically aware, and structured representation space.
>
> - **Analysis on capturing similarity with Sparse representations:**
> To further strengthen our analysis in Figure2 which provides similarity matrices for dense and sparse activations, we defined two clusters within the CIFAR-10 classes: the Animal Cluster, comprising ‘bird,’ ‘cat,’ ‘deer,’ ‘dog,’ ‘frog,’ and ‘horse,’ and the Vehicle Cluster, consisting of ‘airplane,’ ‘automobile,’ ‘ship,’ and ‘truck.’ The comparison of inter- and intra-cluster similarities is visualized in Figure 6 and the results indicate that sparse activations exhibit consistently higher average similarity within clusters compared to dense activations. For the Animal Cluster, sparse activations achieve an average similarity of 0.8873 compared to 0.8367 for dense activations. Similarly, for the Vehicle Cluster, sparse activations achieve 0.8694 compared to 0.8684 for dense activations. This suggests that sparse activations better group objects within the same semantic category, capturing semantic cohesion more effectively. By activating only the most relevant neurons, sparse representations reduce noise and focus on features critical for semantic grouping.
>
> Sparse activations also demonstrate a slightly higher average similarity between clusters, with 0.8295 compared to 0.8068 for dense activations. However, this inter-cluster similarity is significantly lower than the intra-cluster similarity, aligning with the expected semantic distinction between clusters. Notably, sparse representations encode inter-cluster relationships more semantically rather than merely enforcing hard separations. For instance, airplane and bird, which share abstract features like wings and the ability to fly, exhibit a higher similarity of 0.90 with sparse activations compared to 0.87 with dense activations.
>
> Our analysis on the stability and plasticity of the model provides further evidence of the effectiveness of SARL in retaining more information.

---

> ### Author Response · Authors · 2024-11-23
> **Author's Response (2/2)**
>
> > The novelty appears to be somewhat limited, as deep neural networks inherently build and use semantic representations. ... Additionally, an introduction to these losses/similarities should be provided to ensure the reviewer understands they are not being presented as novel contributions.
>
> > In the reviewer's opinion, the paper is a solid combination of well-established techniques that achieves good performance. However, it may be worth considering whether a different venue might be a better fit for this type of contribution.
>
> We respectfully disagree with the reviewer’s assessment that our method lacks novelty or that DNNs inherently capture and retain semantic relationships effectively. While DNNs may implicitly develop semantic representations under static i.i.d. settings (though they still struggle), they fail to retain or leverage these relationships in CL scenarios. In CL, where tasks are learned sequentially, standard neural networks update weights primarily to minimize loss on the current task, often erasing prior knowledge and failing to utilize semantic information from earlier tasks effectively. Even in rehearsal-based methods, which provide limited access to past task samples, the overrepresentation of current task data and the lack of explicit mechanisms to encode and preserve semantic structure create significant challenges.
>
> SARL directly addresses these limitations by introducing a novel framework for semantic-aware learning in CL. Specifically:
>
> - Object Prototypes from Sparse Activations: Unlike standard representations, SARL derives semantically rich object prototypes and utilizes them to inform and retain a semantic structure on the representations.
> - Semantic-Aware Metric Learning: SARL aligns similar representations and separates dissimilar ones across tasks, leveraging the semantic relationships between objects to guide representation learning effectively. This approach is distinct from general metric learning, as it operates within the challenging constraints of CL and focuses on leveraging semantics for forward transfer and interference reduction.
> - Prototype-Based and Functional Regularization: SARL employs prototype-based regularization to maintain the integrity of learned semantic structures and functional regularization to ensure stability during task transitions. These mechanisms collectively mitigate forgetting and reduce recency bias, enabling robust continual learning.
>
> While certain components of our framework, such as consistency regularization with logit replay, build on established techniques, SARL’s novelty lies in its ability to incorporate semantic structure explicitly for continual learning. By leveraging sparse activations and object prototypes, SARL introduces a principled way to encode, retain, and transfer semantic knowledge across sequential tasks.
>
> We would respectfully disagree with the assertion that SARL is merely a combination of well-established techniques. To the best of our knowledge, no existing CL approach explicitly enforces semantic structure in the manner proposed, nor achieves the same degree of performance improvements by leveraging semantic structure in CL scenarios. We are happy to engage further and clarify any perceived overlap with existing methods if the reviewer believes a specific approach is similar to ours.
>
> Finally, regarding venue fit, we believe SARL makes a meaningful contribution to the CL field, particularly with its principled approach to semantic-aware learning. While theoretical analysis is highly valuable, we argue that SARL’s contributions are both innovative and impactful, addressing a critical gap in the literature through comprehensive empirical evidence and practical advancements in continual learning.
>
> We hope our responses address your concerns. We would be happy to enage further, if any concern remains.

---

> > ### Comment · Area_Chair_iJoP · 2024-11-25
> >
> > Dear Reviewer,
> >
> > The authors have provided their responses. Could you please review them and share your feedback?
> >
> > Thank you!

---

> ### Author Response · Authors · 2024-11-29
> **Request for Feedback**
>
> We would greatly appreciate the chance to address any remaining questions or concerns you might have. Please let us know if our initial response addressed your concerns and if we can provide any additional information or clarificantions to address your remaining concerns. During the rebuttal period, we have made significant updates to the manuscript based on all the reviewers’ feedback and suggestions.
>
> Below, we summarize the key changes in the Appendix of the revised manuscript:
>
> - **Evaluation under Online Continual Learning** (Section B.1): Despite not being explicitly designed for Online CL, SARL was adapted to this setting and achieved competitive results compared to state-of-the-art methods. Notably, on Seq-CIFAR100, SARL demonstrated significant gains in both AAA and Acc metrics.
>
> - **Backbone Generalization** (Section B.2): SARL was evaluated on ViT-Small and VGG backbones, demonstrating consistent performance improvements over the ER baseline, highlighting its robustness across architectures.
>
> - **Additional Metrics for Evaluation** (Section C): Beyond average accuracy, we now report metrics such as forgetting, stability, plasticity, and trade-off. These provide a more comprehensive evaluation of SARL’s performance and highlight its balance between knowledge retention and adaptability.
>
> - **Hyperparameter Sensitivity Analysis** (Section D): We conducted a detailed analysis of hyperparameter sensitivity, demonstrating that SARL consistently achieves significant performance gains across a wide range of settings, underscoring its robustness.
>
> - **Extended Analysis of Semantic Representations** (Section E): We compared inter- and intra-cluster similarity matrices for dense and sparse activations, showing that sparse activations better capture meaningful semantic structures.
>
> - **Visualization of Representation Space** (Section F): t-SNE visualizations of object prototypes for SARL and ER illustrate that SARL generates a more structured and semantically meaningful representation space.
>
> We hope these updates strengthen the manuscript and clarify the effectiveness of SARL. If there are specific areas where further clarification is needed, we would be happy to provide additional details. Please let us know if our initial response and these additional results and analysis has addressed your concerns or if there are further improvements you would like us to explore.

---

> > ### Comment · Area_Chair_iJoP · 2024-11-30
> >
> > Dear Reviewer,
> >
> > The authors have provided their responses. Could you please review them and share your feedback?
> >
> > Thank you!

---

> ### Comment · Reviewer_UqLc · 2024-11-30
> **Semantic Interclass Distances in t-SNE: Clarifying Interpretations**
>
> Thank you for sharing the t-SNE visualization. While t-SNE is effective for visualizing local clusters, **inter-cluster distances may not be meaningful**. The reviewer recommends using a scatter plot that directly visualizes original pairwise distances without dimensionality reduction for better accuracy.
>
> If the original dimensionality is high, using a linear layer during learning to reduce it below 10 may help with semantic visualization. Additionally, the analysis in Figure 7 appears incorrect and could benefit from a revision. The reviewer also suggests following the two recommended papers for guidance.

---

> > ### Comment · Reviewer_UqLc · 2024-12-02
> > **Semantic Interclass Distances in t-SNE: Clarifying Interpretations**
> >
> > As a further clarification to my previous comment, it is important to note that t-SNE does not inherently explain inter-cluster distances, which means that there is no semantic-aware structure implied by these distances. While t-SNE is effective for visualizing local clusters, attributing the observed cluster organization to semantic relationships may lead to misinterpretations, as t-SNE primarily preserves local neighbor relationships without providing a reliable basis for understanding inter-cluster distances. For example, the statement that '*animals are tightly grouped together*' cannot be conclusively supported by t-SNE, as it does not accurately reflect semantic grouping between clusters (Appendix F, Fig.7).
> >
> > For these reasons, the concerns raised by the reviewer about the interpretation of the semantic relationships in the representation space may not have been fully addressed.

---

> ### Author Response · Authors · 2024-12-03
> **Clarification on Semantic Interclass Distances**
>
> We sincerely thank the reviewer for their feedback and for highlighting the limitations of t-SNE visualization. We agree that t-SNE is primarily effective for capturing local neighbor relationships and does not inherently reflect global inter-cluster distances. However, we would like to clarify a key aspect of our analysis:
>
> Our t-SNE visualization is not based on individual data samples; rather, it visualizes object prototypes, which are representative vectors for each class in the learned representation space. This distinction is critical because, unlike visualizing individual samples, our analysis focuses on the semantic relationships between classes, as reflected by the organization of their prototypes.
>
> For example, the observation that “bird” and “airplane” are semantically closer is based on the proximity of their respective prototypes, not individual sample clustering. We also encourage the reviewer to consider the t-SNE visualization as a complement to the other analyses provided in the paper. Specifically, we have conducted an extensive analysis using pairwise similarities in the original dimensional space, as shown in Figure 3. These pairwise similarities are consistent with the t-SNE visualization and support the conclusions drawn from it. For instance, in Section E, we analyzed the semantic similarity between clusters in the original dimensionality space, demonstrating that sparse activations enhance both within-cluster and between-cluster semantic similarities.
>
> Building on this, we extended our pairwise similarity analysis to compare the final learned representations of SARL and ER. The results confirm the trends observed in the t-SNE plots. For example:
>
> - The similarity between “bird” and “airplane” is 0.92 for SARL versus 0.87 for ER.
> - The average similarity within the animal cluster is 0.89 for SARL versus 0.87 for ER.
>
> While we are unfortunately unable to revise the manuscript further to include these additional plots, we will incorporate them in the final version to provide a more comprehensive analysis.
>
> Regarding Figure 7, we would appreciate it if the reviewer could elaborate on why the analysis appears incorrect. This would help us refine our interpretation and address any oversights.
>
> We are committed to improving the analysis of representation visualizations based on the feedback provided. Additionally, we would like to inquire whether our overall response, combined with the new empirical results and analyses, sufficiently addresses the reviewer’s initial concerns about the novelty and contribution of our approach.
>
> We would greatly appreciate the opportunity to address any remaining concerns regarding the novelty and contributions of the paper to further increase your confidence in the significance and rigor of our study.

---

### Official Review · Reviewer_oGkj · 2024-11-03

**Soundness:** 3
**Presentation:** 3
**Contribution:** 2
**Rating:** 6
**Confidence:** 4

**Summary:**

This paper tries to strengthen continual learning methods' ability to utilize semantic similarities for efficient learning and knowledge consolidation. The authors propose Semantic Aware Representation Learning (SARL). SARL employs sparse activations and principled approaches to assess similarities between objects encountered in different tasks and subsequently uses them to guide representation learning. Experimental results show that SARL balances plasticity and stability by harnessing underlying semantic structure.

**Strengths:**

The paper is well-structured and easy to follow. The methodology section is thorough and provides detailed explanations.

**Weaknesses:**

1. The method appears to conflict with the problem setting. Specifically, the proposed method stores samples from previous tasks to update the model in future CIL tasks. It seems inconsistent with the CIL problem setting, where samples from previous tasks are not accessible, and storing old samples could lead to information leakage. Please provide a reasonable explanation for this discrepancy.
2. The comparison methods are somewhat limited. The authors only compare their approach with methods that are similar to their own, while overlooking other approaches that also address the CIL problem, such as those based on pre-trained models (e.g., L2P, DualPrompt, etc.). I suggest that the authors clarify their perspective in the “RELATED WORK” or “ANALYSIS” section if they believe these comparisons would not be fair.
3. The evaluations are insufficient. This paper relies solely on “average accuracy” as the evaluation metric, neglecting other important measures such as final accuracy and model forgetting. I recommend including metrics for “top-1 accuracy after the final stage” and “final forgetting” to ensure a comprehensive evaluation of the proposed method.
4. Conflicting notation. Below Eq. (3), the authors state, "... and $C^{0: t}$ represents the set of all object categories observed up to task $t$." Here, "$C^{0: t}$" should be corrected to "$C^{\\{0: t\\}}$". Please carefully proofread the entire paper to ensure consistency in symbols and notation.
5. Typos. In the first paragraph of Appendix A, there is an extra "I" in the last line. Please thoroughly proofread the entire paper to ensure it is free of similar errors.

**Questions:**

Please refer to Strengths and Weaknesses above.

---

> ### Author Response · Authors · 2024-11-22
> **Author's Response (1/3)**
>
> Thank you for your valuable feedback and for recognizing the thoroughness of our methodology and the paper’s structure. We appreciate your comments and suggestions, as they provide us with important insights for improving our work. Below, we address each of your points in detail.
>
> > The method appears to conflict with the problem setting. Specifically, the proposed method stores samples from previous tasks to update the model in future CIL tasks. It seems inconsistent with the CIL problem setting, where samples from previous tasks are not accessible, and storing old samples could lead to information leakage. Please provide a reasonable explanation for this discrepancy.
>
> Thank you for raising this point. We would like to clarify the specifics of the CIL setting and how SARL adheres to well-established practices in this domain. In the CIL setting, the model is tasked with learning a sequence of tasks, each introducing a new set of disjoint classes. The model does not retain access to the full training data from previous tasks, it only leverages a limited memory buffer to store a small subset of samples, a widely accepted strategy in CIL research. This buffer enables models to mitigate catastrophic forgetting without violating the constraints of the CIL setting, as long as the memory usage is consistent across comparisons. Importantly, during inference, SARL does not rely on task identity and operates under the same assumptions as other state-of-the-art methods in the field. [1]
>
> To address catastrophic forgetting in such scenarios, three main categories of approaches have emerged:
>
> - **Regularization-Based Methods**: These methods don't require samples from the previous tasks and instead penalize changes in the model’s parameters or outputs, often utilizing a separate classification head for each task. However, they typically struggle in CIL settings where task information is unavailable, limiting their ability to retain knowledge in this challenging setting.
>
> - **Dynamic Architecture Approaches**: These methods expand the model’s architecture for each task, dedicating specific parameters to avoid forgetting. While effective, the linear increase in model size and the need for task identity during inference limit their scalability and practicality.
>
> - **Rehearsal-Based Methods**: These approaches, which SARL adopts, are inspired by the brain’s experience replay mechanism. A small memory buffer stores a limited number of samples from previous tasks, which are replayed alongside new task samples to approximate the joint distribution. This strategy has been demonstrated as the most general and effective approach across various CL scenarios, particularly in Class-IL settings [2].
>
> SARL adheres to the rehearsal-based approach by using a constrained memory buffer to store a small set of samples. Importantly, this does not conflict with the CIL problem setting. The buffer size is intentionally limited to ensure the model does not have access to full task data from previous stages, preserving the challenges inherent in continual learning. Additionally, our experimental setting makes use of small buffer sizes (200 and 500) and we show that SARL is able to retain information effectively with very limited information. In the Seq-CIFAR100 setting with 200 buffer size, the model only has access to approximately 2 samples per class for the previous tasks.
>
> Rehearsal-based methods like SARL effectively balance stability and plasticity, offering a robust solution to catastrophic forgetting while staying consistent with widely accepted practices in the field. SARL shares the same methodological framework as well-established baselines (e.g., ER, DER++, CLS-ER) in the feld. We hope this explanation addresses your concern, and we are happy to provide further clarification if needed.
>
> **References**:
>
> [1] Van de Ven, Gido M and Tolias, Andreas S, “Three scenarios for continual learning”, {arXiv preprint arXiv:1904.07734, 2019.
>
> [2] Farquhar, Sebastian, and Yarin Gal. "Towards robust evaluations of continual learning." arXiv preprint arXiv:1805.09733 (2018).

---

> ### Author Response · Authors · 2024-11-22
> **Author's Response (2/3)**
>
> > The comparison methods are somewhat limited. The authors only compare their approach with methods that are similar to their own, while overlooking other approaches that also address the CIL problem, such as those based on pre-trained models (e.g., L2P, DualPrompt, etc.). I suggest that the authors clarify their perspective in the “RELATED WORK” or “ANALYSIS” section if they believe these comparisons would not be fair.
>
> SARL is designed to operate in the challenging scenarios where the model starts from a randomly initialized state, without the benefit of pre-training. In contrast, methods like L2P and DualPrompt leverage pre-trained backbones, such as those trained on ImageNet, which provide a substantial advantage in CIL settings. These pre-trained backbones often include representations for a significant portion of the classes encountered later in CL tasks or for classes that are highly similar. As a result, one of the core challenges of CL—learning and preserving general representations of objects across sequential tasks—is largely bypassed in such approaches.
>
> We acknowledge that this distinction is important and have clarified it in the “Extended Related Works” section in the revised manuscript. We emphasize that SARL focuses on settings without pre-training, where the model must build representations from scratch, and outline how this goal differs from prompt-based methods that rely on extensive pre-trained representations. Additionally, while SARL is not currently designed for integration with pre-trained backbones, extending SARL to incorporate such models would be an intriguing direction for future research.
>
> > Conflicting notation and typos
>
> Thank you for pointing out the inconsistency and typos. We apologize for this oversight and will carefully proofread the entire manuscript to ensure consistent notation throughout. We have corrected this specific instance as noted, and proofread the paper for a more polished presentation.

---

> ### Author Response · Authors · 2024-11-22
> **Author's Response (3/3)**
>
> The evaluations are insufficient. This paper relies solely on “average accuracy” as the evaluation metric, neglecting other important measures such as final accuracy and model forgetting. I recommend including metrics for “top-1 accuracy after the final stage” and “final forgetting” to ensure a comprehensive evaluation of the proposed method.
>
> We appreciate your recommendation for additional metrics.
>
>
> We sincerely thank the reviewer for their suggestions regarding evaluation metrics. To address these concerns, we provide a comprehensive evaluation of SARL using multiple metrics, including average accuracy, forgetting, stability, plasticity, and trade-off. Importantly, the average accuracy is equivalent to the top-1 final accuracy in our experiments because the number of test samples is uniform across all tasks for Seq-CIFAR10, Seq-CIFAR100, and Seq-TinyImageNet. This uniformity ensures that both metrics weight task accuracies equally, leading to identical values. We consider
>
> - **Final Forgetting** quantifies the drop in accuracy on a task from the time it is first learned to the end of continual learning training, measuring the model’s ability to retain knowledge.
> - **Stability** reflects the model’s performance on previously learned tasks at the end of training, highlighting how well it preserves earlier knowledge.
> - **Plasticity** measures the model’s ability to learn new tasks, calculated as the average accuracy when tasks are first learned.
> - **Trade-off** evaluates the balance between stability and plasticity, computed as  \text{(2 * Stability * Plasticity) / (Stability + Plasticity)} .
>
> These metrics collectively provide a holistic view of model behavior, capturing the nuances of continual learning scenarios. While the vast majority of baselines do not provide these additional metrics, we include forgetting values (where available) from prior works for comparison. We use the forgetting values from [3] and [4]. Note that DER++ is the strongest single model baseline and SARL considerable reduces forgetting by a considerable margin.}
>
>
> We have added these in the revised manuscript.
>
> **Performance Metrics**:
> | Dataset       | Buffer Size | Avg. Accuracy (%)      | Forgetting (%)      | Stability (%)      | Plasticity (%)      | Trade-off      |
> |---------------|-------------|------------------------|---------------------|--------------------|---------------------|--------------------|
> | seq-cifar10   | 200         | 70.97 ± 0.47          | 14.83 ± 0.43        | 67.72 ± 1.05       | 82.64 ± 0.73        | 74.44 ± 0.78       |
> |               | 500         | 75.64 ± 0.36          | 8.75 ± 6.89         | 73.39 ± 1.00       | 85.31 ± 0.92        | 78.90 ± 0.88       |
> | seq-cifar100  | 200         | 48.96 ± 0.53          | 33.82 ± 0.56        | 43.12 ± 0.91       | 76.45 ± 1.01        | 55.14 ± 0.98       |
> |               | 500         | 55.30 ± 0.61          | 26.78 ± 1.53        | 50.21 ± 1.39       | 76.73 ± 0.62        | 60.69 ± 0.83       |
> | seq-tinyimg   | 200         | 28.95 ± 1.13          | 42.08 ± 2.94        | 23.99 ± 1.41       | 66.82 ± 1.52        | 35.27 ± 1.29       |
> |               | 500         | 32.56 ± 1.23          | 27.34 ± 0.56        | 29.88 ± 1.52       | 57.16 ± 1.73        | 39.24 ± 0.72       |
>
> **Forgetting Comparison**:
> | Buffer | Method   | Seq-CIFAR10  | Seq-CIFAR100  |
> |--------|----------|----------------------------|-----------------------------|
> | 200    | ER       | 61.24 ± 2.62              | 74.16                       |
> |        | FDR      | 84.40 ± 2.67              | -                           |
> |        | DER++    | 32.59 ± 2.32              | 74.05                       |
> |        | SARL     | **14.83 ± 0.43**          | **33.82 ± 0.56**            |
> | 500    | ER       | 45.35 ± 0.07              | 73.64                       |
> |        | FDR      | 85.62 ± 0.36              | -                           |
> |        | DER++    | 22.38 ± 4.41              | 50.54                       |
> |        | SARL     | **8.75 ± 6.89**           | **26.78 ± 1.53**            |
>
> We hope this response addresses your concerns. Your feedback has been extremely valuable in refining our paper, and we look forward to implementing these improvements.
>
>
> **References**:
>
> [3] Buzzega, Pietro, et al. "Dark experience for general continual learning: a strong, simple baseline." Advances in neural information processing systems 33 (2020): 15920-15930.
>
> [4] Boschini, Matteo, et al. "Class-incremental continual learning into the extended der-verse." IEEE transactions on pattern analysis and machine intelligence 45.5 (2022): 5497-5512.

---

> > ### Comment · Area_Chair_iJoP · 2024-11-25
> >
> > Dear Reviewer,
> >
> > The authors have provided their responses. Could you please review them and share your feedback?
> >
> > Thank you!

---

> > ### Comment · Reviewer_oGkj · 2024-11-25
> > **Feedback on Author(s)’ Response**
> >
> > Thank you for the authors' response. Most of my concerns have been addressed, and I have raised my final score from 5 to 6. Below is my detailed feedback:
> >
> > 1. The authors thoroughly explained the CIL problem setup and offered sufficient evidence to demonstrate that replay-based methods are widely recognized as effective approaches for CIL.
> > 2. The authors also clearly explained the use of pre-trained model-based methods and highlighted the key distinctions between the two learning paradigms. These clarifications have been incorporated into the manuscript, and I appreciate the authors' effort in addressing this important aspect.
> > 3. Furthermore, the authors improved the experiments by adding evaluation metrics and comparing the forgetting indicators across different methods, effectively showcasing the proposed approach's performance. While I would have preferred to see additional metrics included for performance comparison, the focus on key methods and metrics provides a solid and thorough evaluation. I find the additional experimental content and results both adequate and satisfactory.

---

> > > ### Author Response · Authors · 2024-12-03
> > > **Comparison with baselines with the additional metrics (1/2)**
> > >
> > > Thank you for your encouraging feedback and for raising your confidence in our study. We sincerely appreciate your recognition of our efforts to address your concerns. Your constructive comments have helped improve the clarity and depth of our work, and we are grateful for your engagement.
> > >
> > > To further incorporate your suggestion of including additional metrics for performance comparison, we trained the strong baselines using the parameters provided in the original paper to calculate these metrics (which were not available in the original works). Specifically, for the exponential moving average (EMA)-based methods (CLSER and SCoMMER), since the EMA model performance is still following the working model at the end of the task learning (see Stable model in FIgure 2 in [1]), we get the highest model performance on each task from working model and subtract the final task performance of the ema model to calculate forgetting. We confirmed this with the authors.
> > >
> > > Below, we present the updated table including the additional metrics.
> > >
> > > | **Dataset**    | **Buffer** | **Method** | **Accuracy (%)**       | **Forgetting (%)**      | **Stability (%)**       | **Plasticity (%)**      | **Trade-off (%)**       |
> > > |-----------------|------------|------------|-------------------------|--------------------------|--------------------------|--------------------------|--------------------------|
> > > | Seq-CIFAR10     | 200        | ER         | 50.36 ± 2.41           | 57.60 ± 3.53            | 38.40 ± 3.01            | **96.44 ± 0.42**        | 54.88 ± 2.98            |
> > > |                 |            | DER++      | 65.74 ± 1.82           | 31.75 ± 2.86            | 58.73 ± 2.53            | 91.13 ± 0.65            | 71.40 ± 1.76            |
> > > |                 |            | CLSER      | 65.92 ± 0.73           | 28.74 ± 1.45            | **67.70 ± 1.21**        | 81.17 ± 1.45            | 73.83 ± 1.21            |
> > > |                 |            | SCoMMER    | 66.80 ± 0.94           | 28.32 ± 1.35            | 62.26 ± 1.49            | 84.72 ± 1.18            | 71.76 ± 0.78            |
> > > |                 |            | SARL       | **70.97 ± 0.47**       | **14.83 ± 0.43**        | 67.72 ± 1.05            | 82.64 ± 0.73            | **74.44 ± 0.78**        |
> > > |                 | 500        | ER         | 62.70 ± 1.04           | 41.92 ± 1.98            | 53.92 ± 1.34            | **96.24 ± 0.86**        | 69.11 ± 1.04            |
> > > |                 |            | DER++      | 70.10 ± 1.47           | 26.79 ± 1.08            | 63.70 ± 1.89            | 91.53 ± 0.64            | 75.11 ± 1.51            |
> > > |                 |            | CLSER      | 75.16 ± 0.86           | 22.56 ± 1.04            | **73.40 ± 0.33**        | 78.03 ± 0.81            | 75.64 ± 0.57            |
> > > |                 |            | SCoMMER    | 74.25 ± 0.42           | 20.13 ± 1.30            | 70.62 ± 0.45            | 87.99 ± 0.45            | 78.36 ± 0.37            |
> > > |                 |            | SARL       | **75.64 ± 0.36**       | **8.75 ± 6.89**         | 73.39 ± 1.00            | 85.31 ± 0.92            | **78.90 ± 0.88**        |
> > > | Seq-CIFAR100    | 200        | ER         | 21.73 ± 0.03           | **76.13 ± 0.29**        | 5.29 ± 0.15             | **82.63 ± 0.21**        | 9.94 ± 0.26             |
> > > |                 |            | DER++      | 30.68 ± 1.35           | 65.10 ± 1.74            | 17.19 ± 1.75            | 82.77 ± 0.72            | 28.44 ± 2.37            |
> > > |                 |            | CLSER      | 43.80 ± 1.89           | 36.33 ± 1.46            | **45.75 ± 1.78**        | 50.47 ± 1.03            | 47.99 ± 1.35            |
> > > |                 |            | SCoMMER    | 40.74 ± 0.53           | 42.20 ± 1.13            | 35.98 ± 0.90            | 63.30 ± 6.21            | 45.80 ± 1.76            |
> > > |                 |            | SARL       | **48.96 ± 0.53**       | 33.82 ± 0.56            | 43.12 ± 0.91            | 76.45 ± 1.01            | **55.14 ± 0.98**        |
> > > |                 | 500        | ER         | 28.03 ± 1.06           | 67.08 ± 1.11            | 13.42 ± 1.60            | **81.69 ± 0.17**        | 23.03 ± 2.37            |
> > > |                 |            | DER++      | 42.03 ± 2.01           | 48.90 ± 2.42            | 32.01 ± 2.74            | 81.15 ± 0.80            | 45.87 ± 2.89            |
> > > |                 |            | CLSER      | 51.68 ± 0.99           | 30.74 ± 0.25            | **52.18 ± 0.27**        | 58.07 ± 2.17            | 54.95 ± 1.04            |
> > > |                 |            | SCoMMER    | 50.11 ± 0.31           | 32.98 ± 0.80            | 46.40 ± 0.79            | 66.93 ± 1.72            | 54.79 ± 0.35            |
> > > |                 |            | SARL       | **55.30 ± 0.61**       | **26.78 ± 1.53**        | 50.21 ± 1.39            | 76.73 ± 0.62            | **60.69 ± 0.83**        |

---

> > > ### Author Response · Authors · 2024-12-03
> > > **Comparison with baselines with the additional metrics (2/2)**
> > >
> > > The results show that SARL considerably reduces forgetting and achieves a superior stability-plasticity tradeoff across all settings.
> > >
> > > We hope these additional analyses further enhance the empirical evaluation and address your suggestion for additional metrics. Please let us know if there are any remaining concerns or analyses we can provide to further increase your confidence in our work.
> > >
> > > [1] Arani, Elahe, Fahad Sarfraz, and Bahram Zonooz. "Learning Fast, Learning Slow: A General Continual Learning Method based on Complementary Learning System." International Conference on Learning Representations.

---

### Official Review · Reviewer_9Ntc · 2024-11-04

**Soundness:** 3
**Presentation:** 2
**Contribution:** 3
**Rating:** 8
**Confidence:** 4

**Summary:**

This paper proposes a novel approach for continual learning by utilizing sparse activations to capture semantic relationships between objects. The authors suggest that leveraging activation sparsity emulates the sparse coding observed in biological neural systems. Prototypes are introduced as a means to represent each object, thereby facilitating knowledge retention across tasks. Experimental evaluations are conducted on benchmark continual learning datasets, including Seq-CIFAR10, Seq-CIFAR100, and Seq-TinyImg, demonstrating the proposed method's performance.

**Strengths:**

- This paper is relatively well-motivated.
- The paper is easy to follow.

**Weaknesses:**

- **Backbone Modifications**: It appears that the backbone architecture is altered (Section 3.3 on Sparse Coding) to incorporate sparse activations, which seem to increase similarities (as shown in Fig. 2). This modification could give the proposed method an advantage over other baselines.

- **Unjustified Claims**:
  - Line 248: The claim that "sparse activations further enhance the model’s ability to distinguish between similar and dissimilar objects" lacks sufficient justification.
  - Line 284: The statement that sparse coding "fosters stability in the model’s knowledge but also mitigates the risk of forgetting important semantic relationships established in earlier learning phases" is unsupported by clear evidence in the text.

- **Inconsistent Notation**:
  - In Equation (3), symbols like \( S_c \) and \( C^t \) should use set notation for clarity.
  - There is inconsistency in interval notation (e.g., \( 0:t \) versus \( \{0:t\} \)).
  - Vectors are not visually distinct from scalars; for example, \( a_i \) and \( o_b^c \) are written in the same style as scalar values, which makes the notation confusing.

- **Objective Complexity and Lack of Ablation**: The objective function comprises five different components, many of which rely on existing objectives, such as consistency regularization. However, no ablation study is provided, making it challenging to determine the contribution of each component to the model's performance:
  - Regularization on object prototypes
  - Consistency regularization loss
  - Semantic-aware metric loss
  - Two cross-entropy losses—one for the current training batch and another for buffer samples

 - **Lack of recent sota baseline**

 [1] Meta continual learning revisited: implicitly enhancing online hessian approximation via variance reduction, ICLR 2024

 [2] Interactive Continual Learning: Fast and Slow Thinking, CVPR 2024

 [3] Bilateral memory consolidation for continual learning, CVPR 2023

**Questions:**

- Given that object representations are \( L_2 \)-normalized, why use the simple mean (Eq. 1) instead of considering a spherical manifold? Why not use similarity (sim) instead of \( L_2 \) distance in Eqs. 4 and 5?

- Some experimental details are missing. For example, the number of steps or epochs for the warm-up stage at each task is not specified. Additional details on the sparse backbone and its comparison with a regular backbone would also be beneficial.

- Memory requirements should be clarified, as different components are stored during training in addition to the buffer from previous tasks (e.g., object prototypes and logits from the previous model state).

---

> ### Author Response · Authors · 2024-11-22
> **Author's Response (1/4)**
>
> Thank you for the valuable feedback. We appreciate your recognition of the paper’s motivation and readability, and we’re glad that you found the motivation behind our approach to be well-articulated. We also appreciate your insights and constructive comments, which have helped us identify areas for further clarification and improvement. We address your concerns below in detail.
>
> > It appears that the backbone architecture is altered (Section 3.3 on Sparse Coding) to incorporate sparse activations, which seem to increase similarities (as shown in Fig. 2). This modification could give the proposed method an advantage over other baselines.
>
> Sparse coding is a fundamental component of our method, inspired by biological neural systems, as discussed in our motivation. The sparse activations aim to emulate neural activity sparsity, fostering enhanced semantic representation by selectively activating the most informative neurons for a given input. To ensure fairness in comparison, we retained the same ResNet-18 backbone across baselines and only replaced ReLU with k-WTA to enforce sparse activations locally. Hence, the number of learning parameters effectively remain the same for all the baselines.
>
> In our ablation study (Table 3), we provide further analysis of sparse activations’ contributions, demonstrating that while it provides a significant performance improvement (supporting our hypothesis), this component alone does not account for the full performance improvement observed with SARL. The remaining improvements are achieved through the integration of semantic-aware metric learning and functional and object prototype regularization.
>
> > Line 248: The claim that "sparse activations further enhance the model’s ability to distinguish between similar and dissimilar objects" lacks sufficient justification.
>
> Thank you for pointing this out, we have added more supporting arguments for this claim in the appendix. To further strengthen our analysis in Figure 2, which provides similarity matrices for dense and sparse activations, we defined two clusters within the CIFAR-10 classes: the Animal Cluster, comprising ‘bird,’ ‘cat,’ ‘deer,’ ‘dog,’ ‘frog,’ and ‘horse,’ and the Vehicle Cluster, consisting of ‘airplane,’ ‘automobile,’ ‘ship,’ and ‘truck.’ The comparison of inter- and intra-cluster similarities is visualized in Figure6 in the Appendix, and the results indicate that sparse activations exhibit consistently higher average similarity within clusters compared to dense activations. For the Animal Cluster, sparse activations achieve an average similarity of 0.8873 compared to 0.8367 for dense activations. Similarly, for the Vehicle Cluster, sparse activations achieve 0.8694 compared to 0.8684 for dense activations. This suggests that sparse activations better group objects within the same semantic category, capturing semantic cohesion more effectively. By activating only the most relevant neurons, sparse representations reduce noise and focus on features critical for semantic grouping.
>
> Sparse activations also demonstrate a slightly higher average similarity between clusters, with 0.8295 compared to 0.8068 for dense activations. However, this inter-cluster similarity is significantly lower than the intra-cluster similarity, aligning with the expected semantic distinction between clusters. Notably, sparse representations encode inter-cluster relationships more semantically rather than merely enforcing hard separations. For instance, airplane and bird, which share abstract features like wings and the ability to fly, exhibit a higher similarity of 0.90 with sparse activations compared to 0.87 with dense activations. In contrast, ship and horse, which are semantically dissimilar, show a similarity of 0.81 with sparse activations compared to 0.83 with dense activations. These findings reinforce our hypothesis that sparse activations can enhance the model's capability to capture semantic structure across objects.
>
> > Inconsistent Notation
>
> Thank you for your attention to detail. We apologize for these inconsistencies, and we have standardized notation in the revised manuscript to improve clarity. We are using the set and vector notation to distinguish.

---

> ### Author Response · Authors · 2024-11-22
> **Author's Response (2/4)**
>
> > The objective function comprises five different components, many of which rely on existing objectives, such as consistency regularization. However, no ablation study is provided, making it challenging to determine the contribution of each component to the model's performance:
>
> We would like to emphasize that we do provide an ablation study in the main manuscript. Section 5.4 and Table 3 provides an ablation study to assess the contribution of the different components on the performance of the model. We show the effect of adding sparse activations, object prototype regularization, semantic aware representation learning and functional regularization. Please let us know if you are looking for a specific combination of components to further assess the contribution of a component. Our ablation study highlights how each term contributes to the performance improvements with SARL, demonstrating that while each component is valuable, they work in synergy and the integration of all components is necessary for achieving optimal performance.
>
> > Lack of recent sota baseline:
> > - [1] Meta continual learning revisited: implicitly enhancing online hessian approximation via variance reduction, ICLR 2024
> > - [2] Interactive Continual Learning: Fast and Slow Thinking, CVPR 2024
> > - [3] Bilateral memory consolidation for continual learning, CVPR 2023
>
> Thank you for highlighting recent works. We appreciate your suggestions and have incorporated these methods into the extended related work section in the appendix, acknowledging their contributions and discussing how they relate to SARL. In our experimental design, we focused on strong rehearsal-based methods as baselines, ensuring a fair comparison by selecting approaches that report results under similar evaluation settings. This includes methods such as CLS-ER and SCoMMER, which incorporate additional models, providing diverse perspectives on continual learning strategies. For methods not evaluated in our experiments, we will clarify any architectural or setting differences that make direct comparison challenging.
>
> We Comparing continual learning methods across diverse baselines presents several challenges. First, evaluation settings differ significantly across works, including variations in architectures, assuming access to explicit or implicit additional information (e.g. the use of LLMs or using pretrained model), CL setting (online vs offline), task structures, and metrics, which complicates direct comparisons. For instance, uniform evaluation settings—such as task definitions, buffer sizes, and training epochs—are critical for meaningful performance assessments. Without a common ground, differences in results may reflect evaluation discrepancies rather than genuine methodological improvements. Therefore, we intentionally adhered to a standardized evaluation protocol to ensure the robustness and reproducibility of our findings.
>
> Regarding the specific methods mentioned by the reviewer:
>
> - [1] is designed for an online continual learning setting, where the model only sees each training example once. While our work focuses on batch training on sequential taks, where models can have multiple passes over the data, leveraging additional training epochs to consolidate knowledge. Given the fundamental differences, direct comparisons are challenging. Similar to many other baselines in our paper, our method requires modifications to be applicable in the online CL. Please let us know if you would still prefer modifying the method for online CL.
>
> - [2] utilizes a multimodal LLM, offering a substantial advantage over SARL and the baselines we have considered. The use of a multimodal LLM introduces an inherent disparity, as such models benefit from extensive pretraining on diverse datasets and significantly larger architectures. We believe a direct comparison would be inherently unfair given these differences. However, we will include a discussion of this work in the related work section to contextualize its contributions.
>
> - [3] is a complementary method that can be integrated with existing approaches, including SARL. We believe this integration could be relatively straightforward and may enhance the performance of SARL further. However, it is important to note that their evaluation setting involves a substantial pretraining phase on a large number of base classes before the sequential learning task. In contrast, SARL and the baselines we compared start continual learning with a randomly initialized model, making the comparison inherently imbalanced. We will clarify this distinction in the manuscript.
>
> We hope this response clarifies the rationale behind the baselines included in our empirical comparison and highlights the challenges of aligning diverse evaluation settings for a fair comparison. As mentioned, we have added rhw discussion of the referred papers in our extended related work.

---

> ### Author Response · Authors · 2024-11-22
> **Author's Response (3/4)**
>
> > Given that object representations are ( L_2 )-normalized, why use the simple mean (Eq. 1) instead of considering a spherical manifold? Why not use similarity (sim) instead of ( L_2 ) distance in Eqs. 4 and 5?
>
> We appreciate this insightful question, as it addresses key design choices in our method. We opted for the simple mean to define class prototypes for several reasons. Firstly, the mean of  L_2 -normalized features inherently lies on the spherical manifold, as the normalization ensures all feature vectors are projected onto the unit sphere. This property aligns with the core objective of capturing semantically meaningful prototypes without introducing additional computational complexity. Thus, the mean provides a computationally efficient and well-aligned representation for continual learning scenarios where computational overhead is a critical consideration. Secondly, using the mean as a prototype representation is a well-established approach in metric learning and object prototype-based methods. The mean effectively captures the central tendency of normalized object representations, summarizing the semantic essence of a class.
>
> Regarding the choice of  L_2 -distance over cosine similarity in Eqs. 4 and 5, we had multiple considerations. While cosine similarity is a viable alternative,  L_2 -distance provides a different degree of penalty for dissimilar points, which can influence the training dynamics. Specifically,  L_2 -distance imposes a stronger penalty for large deviations compared to cosine similarity, which only considers the angular relationship between vectors. This stronger penalty encourages the network to align prototypes more precisely in the feature space.
>
> Your question prompted us to perform additional experiments to analyze the impact of using a spherical manifold and cosine similarity as alternative design choices. Below, we summarize the results of our analysis:
>
> | Prototype Method | Distance Metric | Class-IL Accuracy (%) | Task-IL Accuracy (%) |
> |-------------------|-----------------|------------------------|-----------------------|
> | Mean             | Cosine         | 70.39 ± 1.65          | 95.14 ± 0.33         |
> | Mean             | MSE            | **70.97 ± 0.47**      | **95.62 ± 0.31**     |
> | Sphere           | Cosine         | 70.45 ± 2.22          | 95.36 ± 0.44         |
> | Sphere           | MSE            | 69.51 ± 3.48          | 95.46 ± 0.45         |
>
>
> Using the mean with MSE as the distance metric demonstrates higher consistency, with low variance in accuracy across trials. By contrast, while the sphere-based approach with cosine similarity achieves comparable performancecreating, object prototypes using the sphere-based approach (normalizing by the L2 norm of the mean class representations) exhibits significantly higher variance, particularly in Class-IL accuracy (e.g., 3.48 for Sphere + MSE), which can be problematic for stable performance in continual learning scenarios.
>
> We acknowledge that the sphere-based approach has theoretical appeal, particularly when combined with cosine similarity, as it aligns with normalized feature spaces. However, the observed high variance suggests sensitivity to optimization and data distribution, which may require additional regularization or tuning to stabilize. Overall, while the sphere-based approach with cosine similarity is a valid alternative, our experiments highlight that the mean with MSE provides a robust, computationally efficient, and stable method for defining class prototypes. This balance makes it particularly suitable for continual learning scenarios, where simplicity and consistency are paramount.
>
> > Some experimental details are missing. For example, the number of steps or epochs for the warm-up stage at each task is not specified. Additional details on the sparse backbone and its comparison with a regular backbone would also be beneficial.
>
> Thank you for pointing this out, we apologize for this omission. The warm-up stage is set to 3 epochs for each task in all experiments. We have added this as well as additional details about the sparse backbone in the Appendix. We have also broken down the section for more clarity.
>
> The ablation study in Table 3 shows the effect of using sparse activations (row 2) compared to a dense backbone (row 1). We would like to clarify that the only modification we make to the backbone is replacing the ReLU activation function with the k-winners-take-all (kWTA) mechanism. This change introduces activation sparsity by allowing only the K most active neurons at each spatial location in convolutional layers to propagate, ensuring that the most relevant features are selected locally. This localized sparsity promotes filter specialization, enabling more efficient feature extraction, better semantic encoding, and reduced interference in lifelong learning scenarios. No other architectural changes are made to the backbone.

---

> ### Author Response · Authors · 2024-11-22
> **Author's Response (4/4)**
>
> > Memory requirements should be clarified, as different components are stored during training in addition to the buffer from previous tasks (e.g., object prototypes and logits from the previous model state).
>
> Thank you for the valuable suggestion. SARL maintains an episodic memory for replay (images and logits) and a compact set of object prototypes for efficient retrieval and learning. Additionally, at the beginning of each task, we store the logits for the current task samples to enable functional regularization. Below, we provide a detailed breakdown of the memory requirements for SARL, quantifying each component’s footprint.
>
> For all components, data is stored as 32-bit floating-point values:
> - Memory for 500 images in the buffer: 500 \times 3 \times 32 \times 32 \times 4 bytes = 5.86 MB.
> - Memory for logits of 500 images in the buffer: 500 \times 100 \times 4 bytes = 0.19 MB.
> - Memory for logits at the beginning of the task (functional regularization): 10 \times 500 \times 100 \times 4 bytes = 1.9 MB.
> - Memory for object prototypes (100 classes, 512-dimensional features): 100 \times 512 \times 4 bytes = 0.20 MB.
> - Model size (ResNet-18): 42.66 MB.
>
> As summarized in the table below, the additional memory overhead introduced by SARL compared to Baseline ER is negligible, while providing substantial performance improvements. Furthermore, SARL is significantly more memory-efficient than strong baselines like SCoMMER and CLS-ER, which utilize multiple models and forward passes. When memory efficiency is considered (using the ratio of average accuracy to memory utilization), SARL achieves the best performance among the methods compared.
>
> **Memory and Performance Comparison on Seq-CIFAR100 (500 Buffer Size):**
> | Method   | # Models | # FWD Passes over Task Samples | # FWD Passes over Memory Samples | Memory Buffer Size (MB) | Total Memory (MB) | Memory Ratio (vs ER) | Avg. Acc (%) | Avg. Acc / Memory Ratio |
> |----------|----------|--------------------------------|-----------------------------------|-------------------------|-------------------|-----------------------|--------------|--------------------------|
> | ER       | 1        | 1                              | 1                                 | 5.86                   | 48.52             | 1.000                 | 28.02        | 28.02                   |
> | DER++    | 1        | 1                              | 2                                 | 5.86 + 0.19            | 48.71             | 1.0039                | 41.4         | 41.24                   |
> | CLS-ER   | 3        | 1                              | 3                                 | 5.86                   | 133.84            | 2.7584                | 51.4         | 18.63                   |
> | SCoMMER  | 2        | 1                              | 2                                 | 5.86                   | 91.18             | 1.8792                | 49.63        | 26.41                   |
> | SARL     | 1        | 1                              | 1                                 | 5.86 + 0.19 + 1.9 + 0.20 | 50.09             | 1.0323                | **55.3**     | **53.57**               |
>
> Overall, the additional memory overhead of SARL compared to Baseline ER is minimal, while the accuracy improvements are considerable. SARL also requires significantly less memory than baselines like CLS-ER and SCoMMER, making it a strong, memory-efficient single-model baseline. Moreover, SARL achieves the best Avg. Acc / Memory ratio, underscoring its superior performance when memory utilization is taken into account. We hope these clarifications address your concerns and thank you for your feedback, which has helped us improve the paper. We have also added a section for memory overhead in the Appendix
>
> We hope these responses address your comments and concerns thoroughly. We are grateful for your feedback and for the opportunity to improve our paper.

---

> > ### Comment · Area_Chair_iJoP · 2024-11-25
> >
> > Dear Reviewer,
> >
> > The authors have provided their responses. Could you please review them and share your feedback?
> >
> > Thank you!

---

> > ### Comment · Reviewer_9Ntc · 2024-11-25
> >
> > Thank you for the detailed author response. All my concerns have been addressed, and I am happy to increase my score.
> >
> > The authors clarified my concerns about fairness and the comparison with the baselines. They added a section in the appendix that supports the claim regarding sparse activations, and I agree that this addition strengthens their argument. Their explanations were clear, and I am satisfied with their response. I also thank the authors for the detailed memory analysis and am impressed by the efficiency of SARL.
> >
> > The authors also standardized the notation, making the paper easier to read. Additionally, I am pleased with the details provided regarding the sparse backbone and the number of steps or epochs for the warm-up stage.
> >
> > Regarding the objective ablation: I agree that the three components (OP, SM, FR) are important. However, I was referring to the two cross-entropy losses. If I understand correctly, the two CE losses are always used in Table 3. If this is the case, it would be helpful if the authors clarified this in the text for better understanding.
> >
> > I also agree with the authors that the additional baselines mentioned do not adhere to the same setting as the proposed method and, therefore, are not directly comparable. Nevertheless, I appreciate the extended related work section in the appendix, as well as the comparisons provided in their response.
> >
> > Regarding the claim in the response that the mean of L2-normalized features inherently lies on the spherical manifold: this is not true in general.
> > For example, for v1=[1,0,0] and v2=[0,1,0] the mean of v1 and v2 is [0.5,0.5,0] which have a norm of sqrt(0.5^2+0.5^2)=sqrt(0.5)=0.707 which is not 1.
> > Unless explicitly normalized, the mean vector typically has a norm less than 1, placing it inside the sphere rather than on its surface. The authors provided additional experiments analyzing the impact of using a spherical manifold and cosine similarity. These experiments showed that cosine similarity combined with the spherical manifold is slightly less effective than the (default) Mean + MSE choice but exhibits higher variance.
> >
> > As an additional remark, I think Equation 1 could be replaced with the Fréchet mean (also known as the Riemannian mean) on the sphere. The Fréchet mean minimizes the sum of squared geodesic distances (arccos) to the data points, and Equation 1 could serve as an initialization for this optimization. However, I agree with the authors that this would add complexity, and I am satisfied with their response.

---

> > > ### Author Response · Authors · 2024-11-29
> > > **Author's Reponse (1/2)**
> > >
> > > Thank you for your thoughtful and encouraging feedback on our paper and response. We sincerely appreciate your kind words and detailed suggestions and explanation.
> > >
> > > We are pleased to hear that our explanations and analysis clarified your concerns. The acknowledgment of our memory analysis and the efficiency of SARL, as well as the improved readability from the standardized notation, is highly encouraging.
> > >
> > > Regarding the objective ablation, you are correct that the two cross-entropy (CE) losses are always used in Table 3. We realize this may not have been adequately clarified in the text and we will explicitly mention of this in the final manuscript to ensure better understanding.  Similar to other experience replay-based methods, our approach applies CE loss to both the new task samples and the buffer samples. This ensures that our method remains comparable to the baselines while leveraging the replay mechanism effectively.
> > >
> > > On your remark about the mean of L2-normalized features lying on the spherical manifold, apologoes for the confusion, you are absolutely correct. We appreciate your example demonstrating that the mean vector does not necessarily lie on the sphere’s surface unless explicitly normalized. Additionally, we note your suggestion of using the Fréchet mean on the sphere. While we agree this would add complexity, we also acknowledge its potential to enhance accuracy and will highlighted this in our future work section.

---

> > > ### Author Response · Authors · 2024-11-29
> > > **Author's Response (2/2)**
> > >
> > > **Evaluation under Online Continual Learning Setting** (Added to Appendix Section B.1)
> > >
> > > We also sincerely appreciate your initial suggestion to compare our method with a baseline for online continual learning (Online CL). While this evaluation was initially out of the scope of our study, we recognized its importance and extended our work to include this challenging setting. This exploration not only addresses your recommendation but also helped us uncover the potential of SARL in an entirely new and demanding scenario.
> > >
> > > Online CL is a highly challenging setting where the model is exposed to each data instance only once, requiring it to learn and adapt dynamically without revisiting any data sample. To simulate this scenario, we follow previous works and evaluate our method under the single epoch, single-head setting. Our evaluation includes two key metrics: the final **Averaged Accuracy** (Acc) across all tasks after sequential training and the **Averaged Anytime Accuracy** (AAA) , which measures the model's average performance throughout the training stream.
> > >
> > > Although our method is not specifically designed for Online CL, we adapted it to this setting to evaluate its versatility and applicability. We maintain running feature sums and sample counts to calculate dynamic object prototypes during training, and use the first 10 iterations on a task as warmup before starting the accumulation of features for evaluating prototypes. Despite not being explicitly designed for this setting, SARL-Online demonstrates promising results. Our method achieves improvements across all datasets, particularly on Seq-CIFAR100, where it achieves significant gains in both AAA and Acc.
> > >
> > > | **Method**  | **Seq-CIFAR10** AAA       | **Seq-CIFAR10** Acc       | **Seq-CIFAR100** AAA      | **Seq-CIFAR100** Acc      |
> > > |-------------|---------------------------|---------------------------|---------------------------|---------------------------|
> > > | SGD         | 34.85 ± 1.71             | 16.96 ± 0.60             | 11.63 ± 0.38             | 5.27 ± 0.28              |
> > > | LWF         | 35.31 ± 0.98             | 18.84 ± 0.10             | 11.98 ± 0.40             | 5.63 ± 0.37              |
> > > | A-GEM       | 38.66 ± 0.79             | 18.46 ± 0.17             | 13.15 ± 0.23             | 6.02 ± 0.17              |
> > > | GEM         | 38.67 ± 0.77             | 18.49 ± 0.15             | 15.18 ± 0.38             | 8.30 ± 0.58              |
> > > | ER          | 55.53 ± 2.58             | 43.83 ± 4.84             | 23.19 ± 0.38             | 16.07 ± 0.88             |
> > > | DER         | 45.85 ± 1.62             | 29.87 ± 2.95             | 13.35 ± 0.36             | 6.12 ± 0.18              |
> > > | DER++       | 64.22 ± 0.70             | 52.29 ± 1.86             | 19.88 ± 0.43             | 11.79 ± 0.65             |
> > > | CLSER       | 63.02 ± 1.54             | 52.80 ± 1.66             | 25.46 ± 0.57             | 17.88 ± 0.69             |
> > > | OCM         | 66.14 ± 0.95             | 53.39 ± 1.00             | 22.54 ± 0.79             | 14.40 ± 0.82             |
> > > | ER-OBC      | 65.82 ± 0.91             | 54.85 ± 2.16             | 25.54 ± 0.25             | 17.21 ± 0.92             |
> > > | On-EWC      | 38.44 ± 0.50             | 17.12 ± 0.51             | 11.81 ± 0.42             | 5.88 ± 0.31              |
> > > | IS          | 37.33 ± 0.23             | 17.39 ± 0.19             | 12.32 ± 0.22             | 5.20 ± 0.18              |
> > > | MER         | 50.99 ± 0.65             | 36.92 ± 2.42             | --                        | --                       |
> > > | La-MAML     | 42.98 ± 1.60             | 33.43 ± 1.21             | 12.55 ± 0.39             | 11.78 ± 0.65             |
> > > | VR-MCL      | **66.97 ± 1.58**         | 56.48 ± 1.79             | 27.01 ± 0.48             | 19.49 ± 0.69             |
> > > | **SARL**        | 66.85 ± 1.15             | **57.21 ± 0.27**         | **31.66 ± 1.49**         | **24.39 ± 1.44**         |
> > >
> > >
> > > Additionally, based on another reviewer's suggestion, we also evaluated our method on different backbones (Sppendix Section B.2) to further test the versatility of our approach. We evaluated its performance on the ViT-Small and VGG backbones. Despite the challenges faced by ViT-Small with smaller datasets, SARL consistently provided significant performance gains over the ER baseline across all evaluation settings. The results demonstrate SARL’s robustness and adaptability across diverse architectures. (Please see the general comment on Additional Results for detailed results)
> > >
> > > We would also like to ask if there are additional areas you feel we could explore or clarify further to enhance your confidence in our work. Your suggestions have already helped us strengthen our paper significantly, and we welcome any additional feedback to further improve.
> > >
> > > Thank you again for your time and constructive comments.

---

> > > > ### Author Response · Authors · 2024-12-02
> > > > **Request for feedback**
> > > >
> > > > Thank you for all your feedback and for engaging with us throughout the review process. Your insights and suggestions have been invaluable in helping us strengthen our paper.
> > > >
> > > > As the rebuttal period comes to a close, we wanted to ensure you’ve had a chance to review the additional experiments and analyses we conducted in response to your comments and other reviewer's suggestions. Specifically, we addressed your suggestions with following updates:
> > > >
> > > > - Online Continual Learning (Online CL): Despite not being explicitly designed for Online CL, SARL demonstrated significant gains in both Averaged Accuracy (Acc) and Anytime Accuracy (AAA), particularly on Seq-CIFAR100, showcasing its adaptability and effectiveness in this challenging setting.
> > > >
> > > > - Evaluation on Diverse Backbones: To further validate SARL’s versatility, we tested it on ViT-Small and VGG backbones. SARL consistently outperformed the ER baseline across all settings.
> > > >
> > > > These additional experiments highlight SARL’s robustness and adaptability, addressing your earlier suggestion regarding comparison with additional baselines and further supporting the effectiveness of our method.
> > > >
> > > > If there are any additional areas you feel we could clarify or explore further to increase your confidence in our work, we would be happy to address them.
> > > >
> > > > Thank you again for your thoughtful insights and support.

---

> > > > > ### Comment · Reviewer_9Ntc · 2024-12-03
> > > > >
> > > > > Thank you for providing the additional experiments and updates in response to my suggestions. I appreciate the extra effort you've put into strengthening your paper.
> > > > >
> > > > > Online Continual Learning: It’s encouraging to see that SARL performs well in the Online CL setting without requiring additional memory or replay mechanisms.
> > > > >
> > > > > Evaluation on Diverse Backbones: The evaluation on two relevant backbones and the consistency of the results across them is a valuable addition. I would recommend including an analysis of the impact of sparsification on these backbones as well (Table 3 - Row 1 vs Row 2), as this could provide additional insight into the method.
> > > > >
> > > > > These additional experiments are compelling, and overall, I believe your work is solid and will make a strong contribution to the field. Based on these updates, I am happy to increase my score to 8.

---

> > > > > > ### Author Response · Authors · 2024-12-03
> > > > > > **Acknowledgment and Gratitude**
> > > > > >
> > > > > > Thank you for your thoughtful feedback and for recognizing the additional experiments and updates we incorporated into the paper. Your suggestion to analyze the impact of sparsification across different backbones is highly insightful, and we will include this analysis in the final version of the manuscript to provide further insights into the method’s behavior.
> > > > > >
> > > > > > We deeply appreciate your encouragement and constructive feedback, which have been invaluable in refining our work. We are thrilled by your confidence in our contributions and grateful for your support throughout the review process. We hope that our study will make a meaningful contribution to the field.

---

### Author Response · Authors · 2024-11-23
**Summary of revision**

We sincerely thank the reviewers for their constructive feedback and insightful suggestions, which have significantly improved the clarity and depth of our manuscript. Below, we summarize the key changes made to the revised manuscript in response to the reviewers' comments. All changes are highlighted in blue in the revised document.

1. **Additional Experimental Details (Section A):**
   We have included a more detailed explanation of our experimental settings, including the warm-up stage duration and details of the backbone. This ensures reproducibility and provides a clearer understanding of the methodology.

2. **Additional Metrics for Experimental Settings (Section B, Table 5, and Table 6):**
   Beyond average accuracy, we now report additional metrics such as forgetting, stability, plasticity, and trade-off. These metrics provide a more comprehensive evaluation of SARL's performance, highlighting its strengths in knowledge retention, adaptability, and balance between stability and plasticity.

3. **Extensive Section on Sensitivity to Hyperparameters (Section C, Table 7):**
   We provide results for the complete hyperparameter grid used during tuning, demonstrating that SARL is not overly sensitive to specific parameter values. The method consistently achieves significant performance gains across a broad range of settings, making it practical and robust.

4. **Extension on Similarity of Object Prototypes: Dense vs Sparse Representations (Section D):**
   We extend the analysis of semantic relationships by splitting the CIFAR-10 classes into animals and vehicles clusters comparing the inter and intra custer similarity matrices for dense and sparse activations. This analysis highlights that sparse activations better capture semantic structures.

5. **Visualization of Representation Space Using t-SNE (Section E):**
   We include t-SNE visualizations of the object prototypes for both SARL and ER. These visualizations demonstrate SARL's ability to create a more structured and semantically meaningful representation space, with tighter clustering of related objects and better separation of distinct classes.

6. **Extended Related Work (Section F):**
   We have expanded the related work section to discuss recent advancements, such as meta-CL, multimodal approaches leveraging pretraining, prompting based CL. This contextualization underscores SARL's novelty, particularly its focus on learning representations from scratch without pretraining.

7. **Other Changes:**
   - **Fixed Notations and Typos:** Improved clarity by fixing inconsistencies in mathematical notation and correcting minor typos.
   - **Concise Introduction:** The introduction now more effectively highlights the limitations of existing approaches and succinctly presents SARL's contributions, making the problem statement and novelty more accessible.

These revisions aim to address the reviewers' concerns and provide a more thorough and transparent presentation of our work. We are confident that these updates strengthen the manuscript and demonstrate SARL's significance in advancing continual learning. We deeply appreciate the reviewers' feedback and remain open to further suggestions.

---

### Author Response · Authors · 2024-11-29
**Additional Results (1/3)**

Based on the reviewers’ suggestions, we conducted additional experiments to demonstrate the versatility and applicability of our approach. Below, we summarize the key findings from two significant evaluations. These results have been added to the appendex

**Online Continual Learning** (Section B.1)

Online continual learning (Online CL) is a challenging setting where the model processes each data instance only once without revisiting previous data. To simulate this scenario, we adapted SARL by maintaining dynamic prototypes during training. Despite not being specifically designed for Online CL, SARL demonstrated competitive performance compared to state of the art methods designed for OnlineCL, with significant gains in AAA and Acc metrics, particularly on Seq-CIFAR100.

| **Method**  | **Seq-CIFAR10** AAA       | **Seq-CIFAR10** Acc       | **Seq-CIFAR100** AAA      | **Seq-CIFAR100** Acc      |
|-------------|---------------------------|---------------------------|---------------------------|---------------------------|
| SGD         | 34.85 ± 1.71             | 16.96 ± 0.60             | 11.63 ± 0.38             | 5.27 ± 0.28              |
| LWF         | 35.31 ± 0.98             | 18.84 ± 0.10             | 11.98 ± 0.40             | 5.63 ± 0.37              |
| A-GEM       | 38.66 ± 0.79             | 18.46 ± 0.17             | 13.15 ± 0.23             | 6.02 ± 0.17              |
| GEM         | 38.67 ± 0.77             | 18.49 ± 0.15             | 15.18 ± 0.38             | 8.30 ± 0.58              |
| ER          | 55.53 ± 2.58             | 43.83 ± 4.84             | 23.19 ± 0.38             | 16.07 ± 0.88             |
| DER         | 45.85 ± 1.62             | 29.87 ± 2.95             | 13.35 ± 0.36             | 6.12 ± 0.18              |
| DER++       | 64.22 ± 0.70             | 52.29 ± 1.86             | 19.88 ± 0.43             | 11.79 ± 0.65             |
| CLSER       | 63.02 ± 1.54             | 52.80 ± 1.66             | 25.46 ± 0.57             | 17.88 ± 0.69             |
| OCM         | 66.14 ± 0.95             | 53.39 ± 1.00             | 22.54 ± 0.79             | 14.40 ± 0.82             |
| ER-OBC      | 65.82 ± 0.91             | 54.85 ± 2.16             | 25.54 ± 0.25             | 17.21 ± 0.92             |
| On-EWC      | 38.44 ± 0.50             | 17.12 ± 0.51             | 11.81 ± 0.42             | 5.88 ± 0.31              |
| IS          | 37.33 ± 0.23             | 17.39 ± 0.19             | 12.32 ± 0.22             | 5.20 ± 0.18              |
| MER         | 50.99 ± 0.65             | 36.92 ± 2.42             | --                        | --                       |
| La-MAML     | 42.98 ± 1.60             | 33.43 ± 1.21             | 12.55 ± 0.39             | 11.78 ± 0.65             |
| VR-MCL      | **66.97 ± 1.58**         | 56.48 ± 1.79             | 27.01 ± 0.48             | 19.49 ± 0.69             |
| SARL        | 66.85 ± 1.15             | **57.21 ± 0.27**         | **31.66 ± 1.49**         | **24.39 ± 1.44**         |

---

> ### Author Response · Authors · 2024-11-29
> **Additional Results (2/3)**
>
> **Evaluation on different backbones** (Section B.2)
>
> To further test SARL’s versatility, we evaluated its performance on the ViT-Small and VGG backbones. Despite the challenges faced by ViT-Small with smaller datasets, SARL consistently provided significant performance gains over the ER baseline across all evaluation settings. The results demonstrate SARL’s robustness and adaptability across diverse architectures.
>
> | **Method**  | **Seq-CIFAR10** AAA       | **Seq-CIFAR10** Acc       | **Seq-CIFAR100** AAA      | **Seq-CIFAR100** Acc      |
> |-------------|---------------------------|---------------------------|---------------------------|---------------------------|
> | SGD         | 34.85 ± 1.71             | 16.96 ± 0.60             | 11.63 ± 0.38             | 5.27 ± 0.28              |
> | LWF         | 35.31 ± 0.98             | 18.84 ± 0.10             | 11.98 ± 0.40             | 5.63 ± 0.37              |
> | A-GEM       | 38.66 ± 0.79             | 18.46 ± 0.17             | 13.15 ± 0.23             | 6.02 ± 0.17              |
> | GEM         | 38.67 ± 0.77             | 18.49 ± 0.15             | 15.18 ± 0.38             | 8.30 ± 0.58              |
> | ER          | 55.53 ± 2.58             | 43.83 ± 4.84             | 23.19 ± 0.38             | 16.07 ± 0.88             |
> | DER         | 45.85 ± 1.62             | 29.87 ± 2.95             | 13.35 ± 0.36             | 6.12 ± 0.18              |
> | DER++       | 64.22 ± 0.70             | 52.29 ± 1.86             | 19.88 ± 0.43             | 11.79 ± 0.65             |
> | CLSER       | 63.02 ± 1.54             | 52.80 ± 1.66             | 25.46 ± 0.57             | 17.88 ± 0.69             |
> | OCM         | 66.14 ± 0.95             | 53.39 ± 1.00             | 22.54 ± 0.79             | 14.40 ± 0.82             |
> | ER-OBC      | 65.82 ± 0.91             | 54.85 ± 2.16             | 25.54 ± 0.25             | 17.21 ± 0.92             |
> | On-EWC      | 38.44 ± 0.50             | 17.12 ± 0.51             | 11.81 ± 0.42             | 5.88 ± 0.31              |
> | IS          | 37.33 ± 0.23             | 17.39 ± 0.19             | 12.32 ± 0.22             | 5.20 ± 0.18              |
> | MER         | 50.99 ± 0.65             | 36.92 ± 2.42             | --                        | --                       |
> | La-MAML     | 42.98 ± 1.60             | 33.43 ± 1.21             | 12.55 ± 0.39             | 11.78 ± 0.65             |
> | VR-MCL      | **66.97 ± 1.58**         | 56.48 ± 1.79             | 27.01 ± 0.48             | 19.49 ± 0.69             |
> | SARL        | 66.85 ± 1.15             | **57.21 ± 0.27**         | **31.66 ± 1.49**         | **24.39 ± 1.44**         |
>
> These additional experiments highlight the versatility and robustness of SARL in challenging continual learning scenarios and diverse backbone architectures. We sincerely thank the reviewers for their valuable feedback, which has helped us improve our paper and strengthen the contributions of our work. We believe this further demonstrates SARL’s applicability as a reliable and adaptable continual learning method.

---

> ### Author Response · Authors · 2024-12-03
> **Additional Results (3/3)**
>
> To further incorporate the suggestion of reviwer to include additional metrics for performance comparison, we trained the strong baselines using the parameters provided in the original paper to calculate these metrics (which were not available in the original works). Specifically, for the exponential moving average (EMA)-based methods (CLSER and SCoMMER), since the EMA model performance is still following the working model at the end of the task learning (see Stable model in FIgure 2 in [1]), we get the highest model performance on each task from working model and subtract the final task performance of the ema model to calculate forgetting. We confirmed this with the authors.
>
> The results show that SARL considerably reduces forgetting and achieves a superior stability-plasticity tradeoff across all settings. We provide average and std of three different seeds.
>
> | **Dataset**    | **Buffer** | **Method** | **Accuracy (%)**       | **Forgetting (%)**      | **Stability (%)**       | **Plasticity (%)**      | **Trade-off (%)**       |
> |-----------------|------------|------------|-------------------------|--------------------------|--------------------------|--------------------------|--------------------------|
> | Seq-CIFAR10     | 200        | ER         | 50.36 ± 2.41           | 57.60 ± 3.53            | 38.40 ± 3.01            | **96.44 ± 0.42**        | 54.88 ± 2.98            |
> |                 |            | DER++      | 65.74 ± 1.82           | 31.75 ± 2.86            | 58.73 ± 2.53            | 91.13 ± 0.65            | 71.40 ± 1.76            |
> |                 |            | CLSER      | 65.92 ± 0.73           | 28.74 ± 1.45            | **67.70 ± 1.21**        | 81.17 ± 1.45            | 73.83 ± 1.21            |
> |                 |            | SCoMMER    | 66.80 ± 0.94           | 28.32 ± 1.35            | 62.26 ± 1.49            | 84.72 ± 1.18            | 71.76 ± 0.78            |
> |                 |            | SARL       | **70.97 ± 0.47**       | **14.83 ± 0.43**        | 67.72 ± 1.05            | 82.64 ± 0.73            | **74.44 ± 0.78**        |
> |                 | 500        | ER         | 62.70 ± 1.04           | 41.92 ± 1.98            | 53.92 ± 1.34            | **96.24 ± 0.86**        | 69.11 ± 1.04            |
> |                 |            | DER++      | 70.10 ± 1.47           | 26.79 ± 1.08            | 63.70 ± 1.89            | 91.53 ± 0.64            | 75.11 ± 1.51            |
> |                 |            | CLSER      | 75.16 ± 0.86           | 22.56 ± 1.04            | **73.40 ± 0.33**        | 78.03 ± 0.81            | 75.64 ± 0.57            |
> |                 |            | SCoMMER    | 74.25 ± 0.42           | 20.13 ± 1.30            | 70.62 ± 0.45            | 87.99 ± 0.45            | 78.36 ± 0.37            |
> |                 |            | SARL       | **75.64 ± 0.36**       | **8.75 ± 6.89**         | 73.39 ± 1.00            | 85.31 ± 0.92            | **78.90 ± 0.88**        |
> | Seq-CIFAR100    | 200        | ER         | 21.73 ± 0.03           | **76.13 ± 0.29**        | 5.29 ± 0.15             | **82.63 ± 0.21**        | 9.94 ± 0.26             |
> |                 |            | DER++      | 30.68 ± 1.35           | 65.10 ± 1.74            | 17.19 ± 1.75            | 82.77 ± 0.72            | 28.44 ± 2.37            |
> |                 |            | CLSER      | 43.80 ± 1.89           | 36.33 ± 1.46            | **45.75 ± 1.78**        | 50.47 ± 1.03            | 47.99 ± 1.35            |
> |                 |            | SCoMMER    | 40.74 ± 0.53           | 42.20 ± 1.13            | 35.98 ± 0.90            | 63.30 ± 6.21            | 45.80 ± 1.76            |
> |                 |            | SARL       | **48.96 ± 0.53**       | 33.82 ± 0.56            | 43.12 ± 0.91            | 76.45 ± 1.01            | **55.14 ± 0.98**        |
> |                 | 500        | ER         | 28.03 ± 1.06           | 67.08 ± 1.11            | 13.42 ± 1.60            | **81.69 ± 0.17**        | 23.03 ± 2.37            |
> |                 |            | DER++      | 42.03 ± 2.01           | 48.90 ± 2.42            | 32.01 ± 2.74            | 81.15 ± 0.80            | 45.87 ± 2.89            |
> |                 |            | CLSER      | 51.68 ± 0.99           | 30.74 ± 0.25            | **52.18 ± 0.27**        | 58.07 ± 2.17            | 54.95 ± 1.04            |
> |                 |            | SCoMMER    | 50.11 ± 0.31           | 32.98 ± 0.80            | 46.40 ± 0.79            | 66.93 ± 1.72            | 54.79 ± 0.35            |
> |                 |            | SARL       | **55.30 ± 0.61**       | **26.78 ± 1.53**        | 50.21 ± 1.39            | 76.73 ± 0.62            | **60.69 ± 0.83**        |

---

### Author Response · Authors · 2024-12-03
**Summary of Discussions and Improvements During Rebuttal Period (1/2)**

The rebuttal phase provided valuable opportunities to engage with reviewers and address concerns regarding our study. Below is a concise summary of the key discussions, improvements, and outcomes from this period:

### Key Reviewer Concerns and Responses

### 1. Additional Metrics for Comprehensive Evaluation
- **Concern:** Lack of metrics beyond average accuracy.
- **Response:** Added additional metrics including forgetting, stability, plasticity, and trade-off to provide a holistic view of SARL's performance. Retrained the baselines methods to extend the comparisn with strong baselines. Results demonstrated SARL's superior stability-plasticity trade-off and reduced forgetting across various settings.

### 2. Comparison with additional baselines
- **Concern:** Additional comparisons with baselines including baselines for Online CL, prompt based approaches and methods utilizing a large set of base classes or pretrained models.
- **Response:** Emphasized the importance of evaluating under a uniform setup and the emphasis of our approach on learning robust semantic representations. Extended the releted work section will all the methods shared by the reviewers in the context of our evaluation setting and method. Additionally, though this setting was out of scope for our study and SARL isn't specifically designed for Online CL,  we adapted SARL for the challenging Online CL setting where the model must learn samples with a single pass, achieving competitive results compared to state of art methods designed for Online CL, particularly on **Seq-CIFAR100**, where SARL outperformed existing baseline s on Averaged Anytime Accuracy (AAA) and final accuracy (Acc).

### 3. Evaluation on Additional Backbones
- **Concern:** Experiments relied solely on the ResNet-18 backbone.
- **Response:** Extended evaluations to include **ViT-Small** and **VGG** backbones, showcasing consistent improvements over ER across all settings without hyperparameter tuning.

### 4. Memory and Computational Efficiency
- **Concern:** Potential computational overhead of SARL.
- **Response:** Provided a detailed breakdown of memory usage, showing that SARL achieves significant performance improvements with negligible additional overhead compared to ER. SARL is also significantly more memory and computationally efficient than strong baselines like CLS-ER and SCoMMER.

### 5. Semantic Representation Analysis
- **Concern:** Claims regarding semantic representation learning not sufficiently supported
- **Response:** Added additional analysis on semantic representations using similarity within and between animal and vehicle clusters. Also added t-SNE visualizion of the object  prototypes. Both analysis support SARL's ability to enhance semantic clustering.

### 6. Ablation Studies and Hyperparameter Sensitivity
- **Concern:** Lack of sensitivity analysis on the weights assigned to loss terms.
- **Response:** Added hyperparameter sensitivity experiments, revealing consistent performance across a range of values and highlighting SARL’s robustness.

### 7. Clarifications on the Problem Setup, experimental settings and consistency in notations
- **Concern:** Alignment with the Continual Incremental Learning (CIL) setting and the use of a memory buffer for rehearsal-based methods. Additional details of the backbone and warmsup stage,
- **Response:** Clarified SARL’s adherence to established practices in CIL and contextualized its methodological framework. Extended the experimental setup and improved the consistency in notations


## Key Improvements in the Manuscript

1. **New Experiments:**
   - Evaluations under **Online CL** settings and on diverse backbones (**ViT-Small** and **VGG**) to demonstrate SARL’s versatility and robustness.  (Section B.1, and B.2)
   - Extended analysis of semantic representations using pairwise similarity matrices and t-SNE plot. (Section E and F)
   - Holistic evaluation with additional metrics (forgetting, plasticity, stability, trade-off) (Section C)
   - Hyper parameter sensitivity analysis. (Section D)


2. **Clarifications and Extended Discussions:**
   - Streamlined the introduction to better articulate the limitations of existing methods and SARL's contributions.
   - Expanded related work to include recent state-of-the-art methods, highlighting SARL's distinct position and differences in the evaluation protocol and challenges.(Section G)
   - improved consistency in notations and extended the experimental setup



## Reviewer Feedback and Outcomes

- **Increased Reviewer Scores:**
   - Three reviewers raised their scores from 5 to 6, acknowledging the comprehensive clarifications, additional experiments, and improved clarity in the manuscript.
   - Reviewers appreciated the memory analysis, hyperparameter sensitivity studies, additional metrics and new empirical evaluations.

---

> ### Author Response · Authors · 2024-12-03
> **Summary of Discussions and Improvements During Rebuttal Period (2/2)**
>
> - **Constructive Suggestions for Future Work:**
>    - Exploring alternatives like the Fréchet mean for object prototypes.
>    - Including further analysis of SARL’s approach to leveraging sparse activations and semantic-aware learning, in particular the semantic clustering of object classes.
>
> ## Final Remarks
>
> We sincerely appreciate the reviewers' thoughtful feedback, which guided significant improvements in the manuscript. The additional results and analyses demonstrate SARL's robustness, adaptability, and contributions to lifelong learning. We are committed to addressing any remaining concerns to further enhance confidence in our work.

---

### Meta-Review · Area_Chair_iJoP · 2024-12-16

**Metareview:**

Semantic-Aware Representation Learning (SARL) leverages sparse activations and semantic relationships between tasks to enhance feature reusability, reduce task interference, and achieve a superior balance between plasticity and stability in continual learning.

The paper is well-written. The experiment designs are well-motivated. The experiments are solid and provide a comprehensive analysis of the method.

The paper received mixed reviews with scores of 6, 3, 6, and 8.

Following the rebuttal, the AC summarized a remaining concern raised by reviewer UqLc regarding the t-SNE visualization, noting that it failed to capture inter-cluster distances, and initiated a discussion with the reviewers.

During the internal discussions, both reviewer UqLc and reviewer 9Ntc agreed that the extended pairwise similarity analysis provided stronger support as it directly operated on the prototypes.

Considering the reviewers' agreements and the current status of the paper, the AC decided to accept it. AC strongly encourages the authors to address the reviewers' feedback in the final version, particularly improvements to the t-SNE results and their interpretation.

**Additional Comments On Reviewer Discussion:**

The paper received mixed reviews with scores of 6, 3, 6, and 8.

Following the rebuttal, the AC summarized a remaining concern raised by reviewer UqLc regarding the t-SNE visualization, noting that it failed to capture inter-cluster distances, and initiated a discussion with the reviewers.

During the internal discussions, both reviewer UqLc and reviewer 9Ntc agreed that the extended pairwise similarity analysis provided stronger support as it directly operated on the prototypes.

Considering the reviewers' agreements and the current status of the paper, the AC decided to accept it. AC strongly encourages the authors to address the reviewers' feedback in the final version, particularly improvements to the t-SNE results and their interpretation.

---

### Decision · Program_Chairs · 2025-01-22

Accept (Poster)